# Encoder-Decoder Diffusion Language Models for Efficient Training and Inference

**Marianne Arriola**[*]    **Yair Schiff**[*]    **Hao Phung**    **Aaron Gokaslan**    **Volodymyr Kuleshov**

Department of Computer Science, Cornell University
{marriola,yairschiff,hao}@cs.cornell.edu, {akg87,kuleshov}@cornell.edu
[*] Equal contribution; corresponding authors

## Abstract

Discrete diffusion models enable parallel token sampling for faster inference than autoregressive approaches. However, prior diffusion models use a decoder-only architecture, which requires sampling algorithms that invoke the full network at every denoising step and incur high computational cost. Our key insight is that discrete diffusion models perform two types of computation: 1) representing clean tokens and 2) denoising corrupted tokens, which enables us to use separate modules for each task. We propose an encoder-decoder architecture to accelerate discrete diffusion inference, which relies on an encoder to represent clean tokens and a lightweight decoder to iteratively refine a noised sequence. We also show that this architecture enables faster training of block diffusion models, which partition sequences into blocks for better quality and are commonly used in diffusion language model inference. We introduce a framework for **E**fficient **E**ncoder-**D**ecoder **D**iffusion (E2D2), consisting of an architecture with specialized training and sampling algorithms, and we show that E2D2 achieves superior trade-offs between generation quality and inference throughput on summarization, translation, and mathematical reasoning tasks. We provide the code[1], model weights, and blog post on the project page: https://m-arriola.com/e2d2.

## 1    Introduction

Discrete diffusion models [4], originally proposed as an extension of Gaussian diffusion [28, 59], have steadily improved in quality on tasks, such as language modeling [24, 40, 55], music generation [62], and biological sequence design [56, 68]. Recent successes include diffusion models at the 7-8B parameter scale, which outperform or match comparably sized autoregressive (AR) models [46, 78].

To generate samples, diffusion models iteratively refine a sequence consisting of both clean and corrupted tokens. We observe that this process consists of: 1) producing useful representations of clean tokens and 2) using these representations as context for denoising corrupted tokens. However, prior diffusion language models jointly perform both tasks within the same decoder-only architecture. As a result, these models must invoke the full network at every denoising step, incurring high computational cost.

In this work, we propose an encoder-decoder transformer architecture to separate the computation for representing clean tokens and denoising corrupted tokens. In particular, we use an encoder to represent clean tokens and a lightweight decoder to iteratively refine a noised sequence conditioned on the encoder's representation. This enables faster inference, as we call the lightweight decoder multiple times to iteratively denoise tokens and invoke the encoder only periodically to update its

---

[1]Code: https://github.com/kuleshov-group/e2d2

39th Conference on Neural Information Processing Systems (NeurIPS 2025).

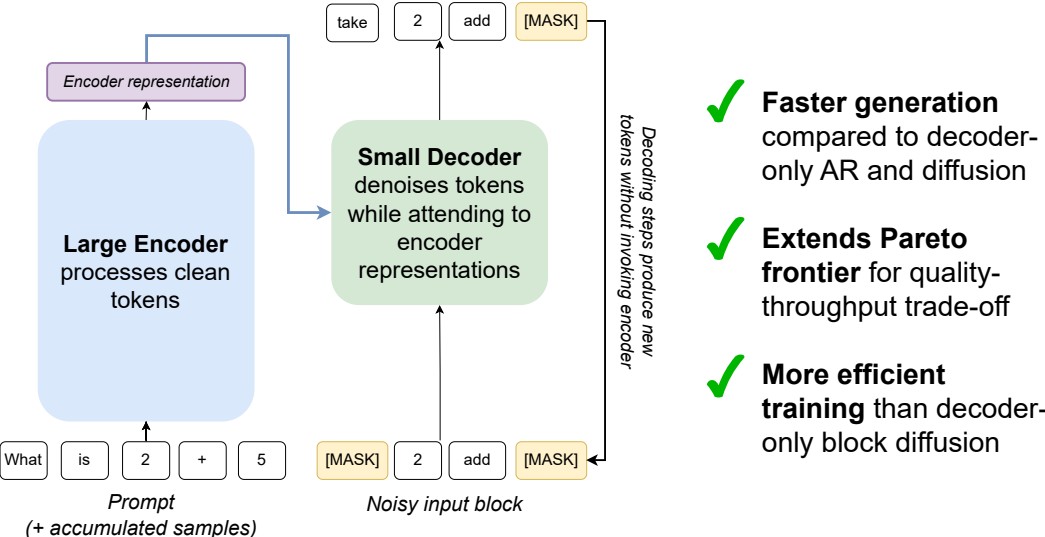

Figure 1: Efficient Encoder-Decoder Diffusion (E2D2) enables faster generation than decoder-only architectures. We accelerate inference by using a lightweight decoder to iteratively denoise for a fixed number of sampling steps, without invoking the encoder. The encoder processes the newly generated tokens periodically to update its representations.

representations. Our novel framework for **E**fficient **E**ncoder-**D**ecoder **D**iffusion (E2D2) consists of an encoder-decoder transformer architecture complemented with efficient training and sampling algorithms that enable both faster inference and KV caching support.

In addition to faster inference, our proposed encoder-decoder architecture enables faster training of block diffusion models [1], which partition sequences into blocks to improve generation quality and support KV caching. Block diffusion is widely used for generating with large diffusion language models, even those trained with standard full-sequence diffusion, such as LLaDA [46], Seed Diffusion [61], MMaDA [77]. However, decoder-only block diffusion incurs higher training costs, with forward passes that are $2\times$ more expensive than standard diffusion, as both the full clean and noised sequences must be processed in every transformer layer [1]. We derive an efficient training algorithm for encoder-decoder block diffusion that uses the encoder to process the clean sequence and the decoder to process the noised sequence, which halves training costs compared to a decoder-only model of equal size.

To demonstrate the efficacy of our encoder-decoder diffusion framework, we train domain-specific diffusion models for summarization, translation, and mathematical reasoning, and show that E2D2 outperforms decoder-only diffusion models in generation speed and quality.

**Contributions** In summary, we make the following contributions: 1) We propose an encoder-decoder architecture for discrete diffusion modeling: the encoder represents clean tokens and the decoder denoises corrupted tokens. 2) We derive efficient diffusion sampling algorithms that accelerate inference by denoising tokens using a lightweight decoder. We also derive an efficient training algorithm for block diffusion modeling using our architecture, which halves training costs compared to a decoder-only architecture. 3) We demonstrate E2D2's effectiveness by training task-specific models on summarization, translation, and mathematical reasoning. We also map the Pareto frontier of the quality-throughput trade-off by varying the decoder size and demonstrate that E2D2 achieves state-of-the-art performance compared to decoder-only discrete diffusion models.

## 2 Background

### 2.1 Discrete Diffusion Models

Diffusion generative models [28, 59, 60] train a denoising network $p_\theta$ to remove noise from latent variables $\mathbf{z}_t$, for $t \in [0, 1]$, obtained from a pre-defined corruption process $q$. This forward process transforms data $\mathbf{x}$ by adding noise increasing in $t$, eventually reaching a limiting distribution $\mathbf{z}_1$.

Discrete denoising diffusion probabilistic models (D3PM [4]) take inspiration from diffusion and define corruption processes over discrete one-hot vectors $\mathbf{z}_t$. A recent successful class of discrete diffusion models focuses on masking (absorbing state) diffusion [47, 55, 58], where the limiting distribution corresponds to $\mathbf{z}_1 = \mathbf{m}$, a one-hot representation of a special [MASK] token. In this framework, the corruption process is defined via interpolation between clean data $\mathbf{x}$ and the prior $\mathbf{m}$: $q(\mathbf{z}_t|\mathbf{x}) = \mathrm{Cat}(\mathbf{z}_t; \alpha_t \mathbf{x} + (1 - \alpha_t)\mathbf{m})$, where $\mathrm{Cat}(\cdot; \cdot)$ represents a categorical distribution and $\alpha_t = \alpha(t) \in [0, 1]$ is a noise schedule decreasing in $t$. For a given corruption process of this form, the true posterior $q(\mathbf{z}_s \mid \mathbf{z}_t, \mathbf{x})$ is known, where $\mathbf{z}_s$ represents less noisy latents for $s < t$.

The optimal diffusion model $p_\theta$ will exactly recover this posterior. However, during inference, we do not have access to $\mathbf{x}$. A common parameterization defines $p_\theta$ as $p_\theta(\mathbf{z}_s|\mathbf{z}_t) = q(\mathbf{z}_s|\mathbf{z}_t, \mathbf{x} = \mathbf{x}_\theta(\mathbf{z}_t))$ where the denoising model $\mathbf{x}_\theta(\mathbf{z}_t)$ predicts clean tokens $\mathbf{x}$ given noisy latents $\mathbf{z}_t$.

To model sequences of $L$ tokens $\mathbf{x}^{1:L}$, the corruption process $q$ is applied independently to each token in a sequence and the denoising network is assumed to factorize independently as well. This framework can easily be extended to accommodate conditional generation by conditioning on some prompt $\mathbf{x}^{1:P}$ in both the forward $q(\mathbf{z}_t^{1:L} \mid \mathbf{x}^{1:L}, \mathbf{x}^{1:P})$ and reverse $p_\theta(\mathbf{z}_s^{1:L} \mid \mathbf{z}_t^{1:L}, \mathbf{x}^{1:P})$ processes.

## 2.2 Block Diffusion Language Modeling

Block diffusion language models (BD3LM [1]) improve sample quality of standard discrete diffusion and enable efficient KV caching by introducing a parameterization that interpolates between AR and fully-parallel diffusion. In block diffusion, blocks of tokens are modeled autoregressively with diffusion applied within each block. Formally, tokens in $\mathbf{x}^{1:L}$ are chunked into $B$ blocks each of length $S$: $B = L/S$. Adopting the shorthand from [1], for each block $b \in [1, B]$, the tokens $\mathbf{x}^{1+(b-1)\cdot S:b\cdot S}$ are denoted as $\mathbf{x}^b$ and all preceding tokens $\mathbf{x}^{1:(b-1)\cdot S}$ are denoted as $\mathbf{x}^{<b}$. The likelihood of this model is defined as $\log p_\theta(\mathbf{x}) = \sum_{b=1}^{B} \log p_\theta(\mathbf{x}^b \mid \mathbf{x}^{<b})$, where each $p_\theta(\mathbf{x}^b \mid \mathbf{x}^{<b})$ is modeled using discrete diffusion. In the reverse process within each block $p_\theta(\mathbf{z}_s^b \mid \mathbf{z}_t^b, \mathbf{x}^{<b})$, we account for already decoded blocks: $\mathbf{x}_\theta = \mathbf{x}_\theta(\mathbf{z}_t^b, \mathbf{x}^{<b})$. As with other diffusion language models, BD3LM uses a decoder-only architecture to parameterize $p_\theta$. See Appendix A for an extended overview.

Models that are trained with standard full-sequence diffusion can apply block decoding at inference by unmasking tokens in blocks left-to-right, such as LLaDA [46], Seed Diffusion [61], MMaDA [77], and others [26, 81, 82]. However, since these models are trained with bidirectional context, they require bidirectional attention over the full sequence, which restricts KV caching and requires approximate caching methods [30, 72, 73]. Training with the block diffusion parameterization described above can further improve sample quality [1, 2, 19].

Naively training a block diffusion model requires a separate function evaluation to predict each block, since the clean context $\mathbf{x}^{<b}$ is different for each block. To enable parallel training along the sequence dimension, [1] concatenate the full clean sequence $\mathbf{x}^{1:B}$ with a noisy sequence $\mathbf{z}_t^{1:B}$. While enabling parallel training, this is difficult to scale since the entire model needs to process $2L$ tokens at every layer, which doubles training cost relative to previous masked diffusion models.

## 3 Efficient Encoder-Decoder Diffusion

Decoder-only transformers [67] are inefficient when used to parameterize $p_\theta$ in language modeling since each inference step requires a full forward pass, motivating architectures that avoid this bottleneck. Our key insight is that discrete diffusion models perform two types of computation: (1) representing clean tokens and (2) denoising masked tokens, which suggests that we can use separate modules for each task.

We explore an encoder-decoder architecture, commonly used in sequence-to-sequence modeling [54, 63, 67], to separate computation used for representing clean tokens and for denoising. Specifically, our probabilistic model $p_\theta$ is defined via the denoising network $\mathbf{x}_\theta$. For this $\mathbf{x}_\theta$, we define 1) an encoder that extracts features from clean tokens, and 2) a decoder that iteratively denoises a sequence of noisy tokens conditioned on these features. Crucially, the decoder can be evaluated multiple times to denoise a sequence of tokens, without needing to invoke the encoder. After a predetermined number of decoder function evaluations, we can then pass newly generated tokens to the encoder, which adds them to its running representation. Unlike existing decoder-only models, our approach

allows parameters and compute to be reallocated between the encoder and decoder (e.g., by using a small decoder and a larger encoder), amortizing the encoder's cost over multiple lightweight decoding steps. We call this approach **E**fficient **E**ncoder-**D**ecoder **D**iffusion (E2D2), consisting of our proposed encoder-decoder architecture complemented with efficient training and sampling algorithms.

We note that E2D2 can be applied to any discrete diffusion parameterization, e.g., both the standard masked diffusion formulation presented in Section 2.1 and the block-wise factorization from Section 2.2. However, in the figures and experimental results, we focus on the block diffusion parameterization for the following reasons: 1) block diffusion attains superior language modeling performance compared to full-sequence masked diffusion [1], 2) block diffusion enables key-value (KV) caching which significantly accelerates inference [1], and 3) even recent state-of-the-art large diffusion language models which are trained with the full-sequence masked diffusion parameterization (e.g., LLaDA [46]) rely on block autoregressive decoding at inference due to its improved performance.

## 3.1 Architecture

**Encoder** We parameterize the encoder with a transformer model and denote the number of layers as $N_{\text{Enc}}$. The input to the encoder is a sequence of clean tokens comprised of both prompt tokens and previously decoded tokens, which we denote as $\mathbf{x}_{t,\text{Enc}}$. The encoder outputs features $\mathbf{h}_t = \text{ENCODER}(\mathbf{x}_{t,\text{Enc}})$. We explore different choices of encoder feature representations in Section 3.2.

**Decoder** As with the encoder, we use a transformer model for the decoder and denote the number of layers as $N_{\text{Dec}}$. The decoder's input, $\mathbf{z}_{t,\text{Dec}}$, corresponds to the portion of the entire sequence $\mathbf{z}_t^{1:L}$ that is "actively being denoised." The decoder outputs logits that are conditioned on the encoder output via cross-attention $\mathbf{x}_{\text{logit}} = \text{DECODER}(\mathbf{z}_{t,\text{Dec}}, \mathbf{h}_t)$, where the positions of $\mathbf{x}_{\text{logit}}$ correspond to $\mathbf{z}_{t,\text{Dec}}$.

## 3.2 Design Decisions

Below, we detail an efficient implementation of the decoder's attention module using kernel fusion. We also explore two ways of connecting the encoder outputs to the decoder: 1) a "last hidden state" version, where the last encoder hidden representation is used as input to the decoder (see left-hand side of Figure 5) and 2) a "shared KV cache" version, where the KV cache is shared between corresponding layers (see right-hand side of Figure 5).

**Fused Attention Kernel** Previous encoder-decoder implementations, e.g., T5 [54], apply self- and cross-attention as separate decoder operations. In contrast, we perform a single fused attention call that jointly attends to both the encoder outputs and the decoder's own hidden states. While this forces the decoder to distribute its attention between the encoder representation $\mathbf{h}_t$ and its own input $\mathbf{z}_{t,\text{Dec}}$, it enables a fused attention kernel that reduces memory access and kernel launch overhead.

**Last Hidden State** The "last hidden state" version follows T5-style encoder-decoder models [54] where we provide the final encoder layer's output features as input to each layer of the decoder. Formally, let $\mathbf{g}_i, \mathbf{f}_i$ be the output of the $i^{\text{th}}$ layer of the encoder and decoder, respectively, with $\mathbf{f}_0 = \text{EMBEDDING}(\mathbf{z}_{t,\text{Dec}})$. Then we define the encoder output as $\mathbf{h}_t = \mathbf{g}_{N_{\text{Enc}}}$. Before each decoder layer, we concatenate this representation to the decoder's input token embeddings. Thus, the input to each decoder layer is $\mathbf{h}_t \oplus \mathbf{f}_{i-1}$, where $\oplus$ denotes concatenation along the sequence dimension.

**Shared KV Cache** The "shared KV cache" variant conditions the decoder on the encoder's intermediate representations by reusing the encoder's KVs from each layer. Unlike the "last hidden state" design where every decoder layer attends to the same $\mathbf{h}_t$, the $i^{\text{th}}$ decoder layer uses the KVs from some $j^{\text{th}}$ encoder layer, where the pairing of $i$ and $j$ is user-specified. This design suits fine-tuning decoder-only models as encoder-decoders, as reusing the encoder's cached KVs keeps the decoder's cross-attention inputs aligned with those expected by the base model and thus improves fine-tuning stability. We define the encoder output as a list of KVs computed per layer $\mathbf{h}_t = [\mathbf{KV}_{\text{Enc}}[1], \ldots, \mathbf{KV}_{\text{Enc}}[N_{\text{Enc}}]]$ where $\mathbf{KV}_{\text{Enc}}[j]$ is the KV cache of the $j^{\text{th}}$ encoder layer. Let $\mathbf{KV}_{\text{Dec}}[i]$ be the KVs computed at the $i^{\text{th}}$ decoder layer using the output from the previous decoder layer. Then, the keys and values used for the fused cross- and self-attention in the decoder are formed via the concatenation $\mathbf{h}_t[j] \oplus \mathbf{KV}_{\text{Dec}}[i]$.

## 3.3 Sampling Algorithm

E2D2 models admit faster inference by decoupling encoder and decoder calls. Whereas prior decoder-only language models require calling the full transformer in every generation step, E2D2

calls a lightweight decoder multiple times throughout denoising, followed by one call to a larger encoder. In Algorithm 1, we present the sampling procedure for E2D2 applied to the block diffusion parameterization, which generates block-by-block while enabling efficient KV caching. We begin by embedding the context or prompt of $P$ tokens through the encoder to produce encoder features $\mathbf{h}^{1:P}$, and we pre-fill the encoder KV cache. We generate a new block by iteratively denoising a block of masked tokens conditioned on $\mathbf{h}^{1:P}$, invoking only the decoder.

We adjust the encoder's signature to accept the KV cache: $\text{ENCODER}(\mathbf{x}^b, \mathbf{KV}^{<b})$, where $\mathbf{KV}^{<b}$ represents the KV cache for all blocks preceding $b$. We also adjust the output of the encoder so that it returns the updated KV cache for the newest block $\mathbf{KV}^b$. For the "last hidden state" version of E2D2, which computes separate KVs for the encoder and decoder, we also accumulate a KV cache for the decoder (not currently depicted in Algorithm 1).

---

**Algorithm 1** Encoder-Decoder Block Diffusion Sampling with KV Caching

---

**Require:** Prompt $\mathbf{x}^{1:P}$, # of blocks $B$, # of diffusion steps $T$, ENCODER, DECODER, algorithm SAMPLE
1:  $\mathbf{x}^{1:B}, \mathbf{KV} \leftarrow \emptyset, \emptyset$                                                    ▷ Output, KV cache
2:  $\mathbf{h}, \mathbf{KV} \leftarrow \text{ENCODER}(\mathbf{x}^{1:P}, \emptyset)$                   ▷ Encode prompt, collect encoder KV cache
3:  **for** $b = 1$ to $B$ **do**
4:      $\mathbf{z}_t^b \sim \mathbf{m}^b$                                          ▷ Initialize noised block
5:      **for** $t = 1, \frac{T-1}{T}, \ldots \frac{1}{T}$ **do**
6:          $\mathbf{z}_t^b \leftarrow \text{SAMPLE}(\text{DECODER}(\mathbf{z}_t^b, \mathbf{h}))$        ▷ Denoise block: Apply small decoder only
7:      **end for**
8:      $\mathbf{x} \leftarrow \mathbf{x}^{<b} \oplus \mathbf{z}_t^b$                               ▷ Accumulate output
9:      $\mathbf{h}, \mathbf{KV}^b \leftarrow \text{ENCODER}(\mathbf{x}^b, \mathbf{KV}^{<b})$       ▷ Encode new block, compute encoder KVs
10:     $\mathbf{KV} \leftarrow \mathbf{KV}^{<b} \oplus \mathbf{KV}^b$                     ▷ Accumulate KV cache
11: **end for**
12: **return** $\mathbf{x}^{1:B}$

---

Table 1: Forward-pass training FLOPs comparison. $N$ denotes the number of layers, $L$ the sequence length, $S$ the block size, and $D$ the hidden dimension. The expression highlighted in blue denotes the number of attention operations. E2D2 refers to block diffusion parameterization. For E2D2, we separate $N$ into $N_{\text{Enc}}$ encoder and $N_{\text{Dec}}$ decoder layers. We provide the derivation in Appendix B.

Table 2: Inference FLOPs comparison. $O(\theta)$ is cost of a decoder call and $O(\phi)$ is cost of an encoder call. For E2D2, $O(\theta) \ll O(\phi)$. $T$ denotes diffusion steps and $B$ the number of blocks. E2D2 uses the block diffusion parameterization.

| | Attention FLOPs | MLP FLOPs |
|---|---|---|
| AR | $4ND(\frac{L^2+L}{2})$ | $24NLD^2$ |
| MDLM | $4NDL^2$ | $24NLD^2$ |
| BD3LM | $4ND(L^2 + LS)$ | $48NLD^2$ |
| E2D2 | $4(N_{\text{Enc}} + N_{\text{Dec}})D(\frac{L^2+LS}{2})$ | $24(N_{\text{Enc}} + N_{\text{Dec}})LD^2$ |

| | Inference FLOPs |
|---|---|
| AR | $L \cdot O(\theta)$ |
| MDLM | $T \cdot O(\theta)$ |
| BD3LM | $BT \cdot O(\theta)$ |
| E2D2 | $\underbrace{B \cdot O(\phi)}_{\text{encoder}} + \underbrace{BT \cdot O(\theta)}_{\text{decoder}}$ |

## 3.4 Training Algorithm

In this section, we review the algorithm and complexity of a forward pass through the E2D2 model to compute the loss of the block diffusion objective. The training procedure for E2D2 is presented in Algorithm 2. We model the posterior probabilities $p_\theta(\mathbf{z}_s^b | \mathbf{z}_t^b, \mathbf{x}^C)$ for all blocks $b \in [1, \ldots, B]$ and context tokens $\mathbf{x}^C$, where $C \subset \{1, \ldots, L\}$ is a set of their position indices. For block diffusion, context tokens for block $b$ correspond to previous clean blocks $\mathbf{x}^C = \mathbf{x}^{<b}$. Naively, we would compute logits by invoking the encoder and decoder for each block in a loop, since the logits for each block are conditioned on block-specific inputs $\mathbf{z}_t^b, \mathbf{x}^C$. This entails calling the encoder $\mathbf{h}_t^C = \text{ENCODER}(\mathbf{x}^C)$ to encode the conditioning $\mathbf{x}^C$ and the decoder to compute $\mathbf{x}_{\text{logit}}^b = \text{DECODER}(\mathbf{z}_t^b, \mathbf{h}_t^C)$.

**Vectorized Implementation** To efficiently model all posterior probabilities across $B$ blocks using a single encoder and decoder pass, we process all clean tokens $\mathbf{x}^{1:L}$ through the encoder and noised tokens $\mathbf{z}_t^{1:L}$ through the decoder. We design custom attention masks (Figure 2) to ensure that noisy tokens attend within their noised block and to previous clean blocks, inspired by [1].

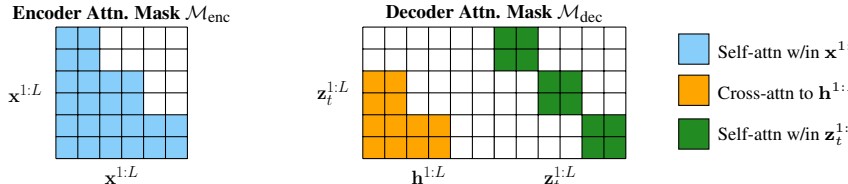

Figure 2: Example attention masks used in block diffusion training with an encoder-decoder architecture, for $L = 6$ tokens with blocks of size $S = 2$. **Left:** The encoder mask $\mathcal{M}_{\text{Enc}} \in \{0,1\}^{L \times L}$ enables clean tokens to attend within their respective block and to previous blocks. **Right:** The decoder mask $\mathcal{M}_{\text{Dec}} \in \{0,1\}^{L \times 2L}$ uses self-attention within noised blocks and cross-attention to previous clean blocks using the encoder's output representation $\mathbf{h}^{1:L}$.

In particular, the clean sequence $\mathbf{x}^{1:L}$ is passed to the encoder with a block-causal mask $\mathcal{M}_{\text{Enc}} \in \{0,1\}^{L \times L}$ to produce the representations $\mathbf{h}^{1:L}$ so that each clean block is conditioned on itself $\mathbf{x}^b$ and previous blocks $\mathbf{x}^{<b}$(Figure 2, left). In the decoder, for a given noisy block $\mathbf{z}_t^b$, each token attends to tokens within its block $\mathbf{z}_t^b$ and clean token representations from preceding blocks $\mathbf{x}^{<b}$, which we denote as $\mathbf{h}^{<b}$. The decoder attention mechanism operates over $2L$ keys and values corresponding to the $L$ tokens in $\mathbf{h}^{1:L}$ and the $L$ to-

---

**Algorithm 2** Encoder-Decoder Training

**Require:** datapoint $\mathbf{x}^{1:L}$, noise process $q_t$, ENCODER w/ params. $\phi$, DECODER w/ params. $\theta$, diffusion loss $\mathcal{L}$
1: **repeat**
2:     Sample $t \sim \mathcal{U}[0,1]$
3:     $\mathbf{z}_t^{1:L} \sim q_t(\cdot|\mathbf{x}^{1:L})$
4:     $\mathbf{h}^{1:L} \leftarrow \text{ENCODER}(\mathbf{x}^{1:L}, \mathcal{M}_{\text{Enc}})$
5:     $\mathbf{x}_{\text{logit}}^{1:L} \leftarrow \text{DECODER}(\mathbf{z}_t^{1:L}, \mathbf{h}^{1:L}, \mathcal{M}_{\text{Dec}})$
6:     Take gradient step on $\nabla_{\theta,\phi}\mathcal{L}(\mathbf{x}_{\text{logit}}^{1:L}; \theta, \phi)$
7: **until** converged

---

kens in $\mathbf{z}_t^{1:L}$. The decoder attention pattern is enforced through an attention mask $\mathcal{M}_{\text{Dec}} \in \{0,1\}^{L \times 2L}$ (Figure 2, right). We thus perform an encoder pass over the entire clean sequence, followed by a decoder pass over the entire noisy sequence, as follows: $\mathbf{h}^{1:L} = \text{ENCODER}(\mathbf{x}^{1:L}, \mathcal{M}_{\text{Enc}})$ and $\mathbf{x}_{\text{logit}}^{1:L} = \text{DECODER}(\mathbf{z}_t^{1:L}, \mathbf{h}^{1:L}, \mathcal{M}_{\text{Dec}})$.

Crucially, using a smaller decoder significantly improves training throughput relative to a decoder-only BD3LM model. In Table 1, we depict the FLOPs comparison between modeling paradigms and E2D2 applied to block diffusion, see Appendix B for a detailed derivation. Letting $N$ be the total number of layers in the model, which for E2D2 we have $N = N_{\text{Enc}} + N_{\text{Dec}}$, we see that for same number of layers, E2D2 uses $2\times$ fewer FLOPs compared to BD3LM. Moreover, for any block diffusion model of block size $S < L$, we achieve fewer training FLOPs relative to standard masked diffusion models (MDLM; see Appendix C) while maintaining the superior quality of the BD3LM block diffusion parameterization.

### 3.5 E2D2 for Standard Masked Diffusion (MDLM)

Below, we present details for training and sampling from a standard full-sequence masked diffusion model (MDLM [55]), introduced in Section 2.1, using our encoder-decoder architecture. Recall that here our denoising network predicts a joint distribution over the entire sequence $p_\theta(\mathbf{z}_s^{1:L} \mid \mathbf{z}_t^{1:L})$, unlike block diffusion which requires predicting the posterior for each block of tokens.

The encoder takes as input a sequence of clean context tokens consisting of prompt tokens and clean tokens in $\mathbf{z}_t^{1:L}$. The decoder takes as input the full noised sequence $\mathbf{z}_t^{1:L}$, which outputs logits for each token conditioned on the encoder's output. This design facilitates faster inference, as the lightweight decoder may be invoked multiple times to generate a number of tokens before calling the larger encoder to produce a strong contextual representation.

**Sampling** For full-sequence masked diffusion, the sampling procedure follows Algorithm 1, except that here the decoder generates groups of tokens that can be in any position of the full sequence rather than in contiguous blocks. For a more formal exposition see Appendix D and Algorithm 3. As above, we begin by encoding the $P$ prompt tokens $\mathbf{x}^{1:P}$ to attain the encoder representation $\mathbf{h}^{1:P}$. The decoder's input is initialized to a sequence of length $L$ consisting of only masked tokens. The decoder iteratively denoises this full sequence for a fixed number of steps with cross-attention to $\mathbf{h}^{1:P}$. At the end of this denoising interval, the decoder has generated some new tokens, which are potentially

non-contiguous in the sequence dimension. These tokens are returned as a new 'block' to the encoder, which adds them to its running context and produces an updated clean token representation. This process is repeated until the sampling budget $T$ is exhausted.

**Training** At every training step, we sample $\mathbf{z}_t^{1:L} \sim q(\mathbf{z}_t^{1:L} \mid \mathbf{x}^{1:L})$. The input to the encoder consists of the clean tokens in this sequence. The decoder receives the entire sequence $\mathbf{z}_t^{1:L}$ and the encoder's representation of the clean tokens. Since we model the full sequence with diffusion, both the encoder and decoder use bidirectional attention. See Appendix D for more detail.

## 4 Experiments

### 4.1 Experimental Design

We study the capabilities of E2D2 on summarization, translation, mathematical reasoning, and zero-shot likelihood estimation on held-out data. In each setting, we train domain-specific E2D2 and baseline models. In addition to performance metrics for each task, we report inference throughput. By varying the decoder size, we can map the Pareto frontier of efficiency and performance, with increasing number of layers controlling both performance (larger decoder $\rightarrow$ higher performance) and throughput (larger decoder $\rightarrow$ lower throughput). As noted in Section 3, throughout the experiments, we use the block diffusion parameterization in conjunction with E2D2.

**Datasets** We examine 1) text summarization (CNN/DailyMail; [27, 57]) for which we compute ROUGE scores [37], 2) machine translation (WMT 14 de-en; [6]) for which we compute the BLEU [50] score, and 3) mathematical reasoning (GSM8K; [10]) for which we compute zero-shot pass@1 accuracy. We also train E2D2 on the widely used pretraining OpenWebText dataset [21]. We compute perplexity (PPL) on the validation set of this corpus and PPL for held-out datasets (zero-shot PPL).

**Baselines** We compare E2D2 against decoder-only AR models and discrete diffusion models MDLM [55] and BD3LM [1]. For likelihood estimation, we also compare against SEDD [40].

**Last Hidden State Design Decisions** For the last hidden state version of E2D2, models are parameterized using a transformer architecture based on Qwen3 [76]. Qwen3 features improvements over prior architectures, such as removing QKV biases and introducing QK normalization [14]. We maintain untied weights for the encoder and decoder. For the CNN/DailyMail, WMT, and OWT datasets, we train from scratch using this last hidden state version of E2D2. Full model and training hyperparameters are detailed in Appendix E.

**Shared KV Cache Design Decisions** For the shared KV cache variant of E2D2, we use pretrained Qwen3 weights to instantiate an E2D2 encoder, letting $N_{\text{Enc}}$ be the same as the underlying Qwen3 LLM. We then copy these weights for the decoder, but reduce the number of layers, to instantiate a lightweight decoder so that $N_{\text{Dec}} < N_{\text{Enc}}$. In particular, we copy layers closer to the output. The KV cache from the encoder layer is provided to the decoder layer from which it was copied, i.e., at the $i^{\text{th}}$ decoder layer we use $\mathbf{KV}[N_{\text{Enc}} - N_{\text{Dec}} + i]$. We reduce the memory footprint of the model by weight-tying the encoder and decoder parameters.

For GSM8K, we fine-tuned from a pretrained Qwen3 1.7B model using this shared KV cache architecture. Full details are in Appendix E. We find that for tasks where the downstream dataset is relatively small, e.g., GSM8K, initializing from the pretrained weights of the underlying Qwen model improves convergence and downstream performance. This is similar to previous findings that initializing diffusion training from a pretrained AR model improves convergence [22, 78].

### 4.2 Results

**Summarization and Machine Translation** For text summarization and machine translation, in Tables 3 and 4, we see that E2D2 is able to match or outperform our diffusion baselines while achieving higher throughput. On the summarization task, E2D2 even outperforms an AR baseline that has the same number of layers, while increasing throughput by $\sim$75%. We find that AR overfits on this dataset, unlike diffusion, which is robust against overfitting by training over masking rates [43, 52]. Compared to MDLM, which does not support KV caching, E2D2 offers better language modeling and downstream task performance with $\sim$3$\times$ faster inference. In Table 4, E2D2 achieves higher throughput and better BLEU scores for translation relative to the 16-layer BD3LM. While a small 12-layer BD3LM approaches the throughput of E2D2, its BLEU score worsens further.

Table 3: CNN/DailyMail test set ROUGE scores. Best values for our trained models are **bolded.** $N$ refers to number of transformer layers ($N_{Enc}/N_{Dec}$ for E2D2). Decoding throughput (`Tput`) is measured in tokens / sec on 1 `H100 80GB` machine. For all models, we use $T = L$ sampling steps, so the throughput can be higher for diffusion when $T < L$. We report mean $\pm$ standard deviation for 100 samples.

|  |  |  | ROUGE ($\uparrow$) | | |
|  |  |  | 1 | 2 | L |
|---|---|---|---|---|---|
| *Past baselines* |  |  |  |  |  |
| GPT-2 [53] |  |  | 29.3 | 8.3 | 26.6 |
| BERT-L [39] |  |  | 41.7 | 19.4 | 38.8 |
| T5-L [54] |  |  | 42.5 | 20.7 | 39.8 |
| AR-Diff. ($k = 50$) [74] |  |  | 39.6 | 16.3 | 37.1 |
| GENIE ($k = 50$) [38] |  |  | 29.3 | 8.3 | 21.9 |
| *From scratch* | $N$ | `Tput` ($\uparrow$) | 1 | 2 | L |
| AR | 28 | $89.1_{\pm0.6}$ | 31.7 | 11.7 | 22.1 |
| MDLM | 28 | $49.3_{\pm14.6}$ | 30.6 | 12.5 | 22.7 |
| BD3LM | 12 | $135.1_{\pm3.3}$ | 35.8 | 13.7 | 23.7 |
| E2D2 (Ours) | 20/8 | $\mathbf{155.8}_{\pm2.6}$ | **36.0** | **14.1** | **23.9** |

Table 4: WMT (*de-en*) test set BLEU score. Best values for our trained models are **bolded.** $N$ refers to number of transformer layers ($N_{Enc}/N_{Dec}$ for E2D2). Decoding throughput (`Tput`) is measured in tokens / sec on 1 `H100 80GB` machine. For all models, we use $T = L$ sampling steps, so the throughput can be higher for diffusion when $T < L$. We report mean $\pm$ standard deviation for 100 samples.

|  |  |  | BLEU ($\uparrow$) |
|---|---|---|---|
| *Past baselines* |  |  |  |
| NAT [23] |  |  | 20.62 |
| NAR autoencoders [34] |  |  | 25.43 |
| USMT [3] |  |  | 17.43 |
| CMLM Base [20] |  |  | 29.47 |
| *From scratch* | $N$ | `Tput` ($\uparrow$) | BLEU ($\uparrow$) |
| AR | 32 | $77.6_{\pm0.4}$ | **25.2** |
| MDLM | 32 | $60.4_{\pm0.8}$ | 18.4 |
| BD3LM | 12 | $129.6_{\pm0.7}$ | 23.3 |
| BD3LM | 16 | $102.4_{\pm0.5}$ | 24.0 |
| E2D2 (Ours) | 28/4 | $\mathbf{162.0}_{\pm1.4}$ | 24.8 |

Table 5: Evaluation on GSM8K test set. Best diffusion value is **bolded**. $N$ refers to number of transformer layers ($N_{Enc}/N_{Dec}$ in the case of E2D2). Decoding throughput (`Tput`) is measured in tokens / sec on 1 `H100 80GB` machine. For all models, we use $T = L$ sampling steps, so the throughput can be higher for diffusion when $T < L$. We report mean $\pm$ standard deviation for 100 samples.

|  | $N$ | PPL ($\downarrow$) | 0-shot pass@1 ($\uparrow$) | `Tput` ($\uparrow$) |
|---|---|---|---|---|
| *Fine-tuned* |  |  |  |  |
| AR | 28 | 1.49 | 66.6 | $94.1_{\pm0.5}$ |
| MDLM | 28 | $\leq 2.30$ | 14.0 | $31.9_{\pm3.0}$ |
| BD3LM | 21 | $\leq 1.87$ | 33.2 | $86.6_{\pm0.5}$ |
| E2D2 (Ours) | 28/14 | $\leq \mathbf{1.80}$ | **47.9** | $\mathbf{102.8}_{\pm0.6}$ |

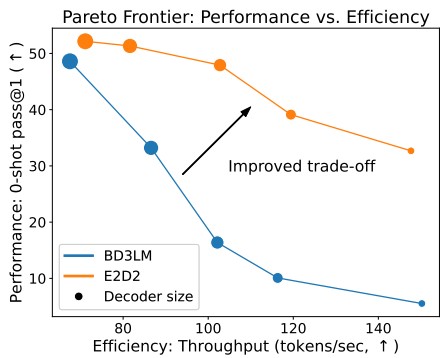

Figure 3: Mapping the Pareto Frontier: Larger models increase accuracy on GSM8K at the cost of slower decoding. E2D2 improves this trade-off.

**Mathematical Reasoning** Table 5 contains results on the GSM8K benchmark. As above, E2D2 shows improved downstream performance and decoding throughput compared to diffusion baselines. Qualitatively, we also observe that E2D2 outperforms BD3LM models; see samples in Appendix G.

**Mapping the Pareto Frontier of Performance and Efficiency** By varying the depth of E2D2's decoder and that of baseline models, we can examine the trade-off between performance (which increases with larger models) and throughput (which decreases with larger models). We fine-tune models on the GSM8K dataset and compute 0-shot pass@1 accuracy and decoding throughput. We select the number of decoder layers to roughly match the throughput for E2D2 and BD3LM models at various sizes, with BD3LM decoder layers varying over $N \in \{10, 14, 17, 21, 28\}$ and E2D2 (using $N_{Enc} = 28$) decoder layers varying over $N_{Dec} \in \{6, 10, 14, 21, 26\}$. At each throughput level, E2D2 features higher quality, extending the Pareto frontier of quality and speed in Figure 3.

**Likelihood** In addition to downstream task performance, we evaluate E2D2's language modeling capabilities. We train E2D2 model on OpenWebText (OWT; [21]) and compute zero-shot PPL ($\downarrow$) on held-out benchmark datasets from [53]. E2D2 with the block diffusion parameterization outperforms

full-sequence diffusion baselines SEDD and MDLM on OWT perplexity. While attaining comparable perplexities to BD3LM, we find that E2D2 is 40% faster to train, as measured in training throughput (tokens / sec on 1 `H100 80GB`, with batch size 32 and context length $L = 1024$, measured after 100k steps) with a throughput of $8.4 \times 10^4$ for E2D2 and $5.9 \times 10^4$ for BD3LM.

Table 6: Validation perplexities ($\downarrow$) for models trained on OWT. Perplexities for diffusion models are upper bounds. All models are trained on 524B tokens. All models use 170M total parameters and $N = 12$ total layers; for E2D2, we use $N_{\text{Enc}} = 10, N_{\text{Dec}} = 2$. † indicates values taken from [1].

|  | OWT | PTB | Wikitext | LM1B | *Zero-shot*
Lambada | AG News | Pubmed | Arxiv |
|---|---|---|---|---|---|---|---|---|
| AR† | 17.54 | 81.07 | 25.32 | 51.14 | 52.13 | 52.11 | 48.59 | 41.22 |
| SEDD† | 24.10 | 96.33 | 35.98 | 68.14 | 48.93 | 67.82 | 45.39 | 40.03 |
| MDLM† | 22.98 | 90.96 | 33.22 | 64.94 | 48.29 | 62.78 | 43.13 | 37.89 |
| BD3LM† | 20.73 | 96.81 | 31.31 | 60.88 | 50.03 | 61.67 | 42.52 | 39.20 |
| E2D2 | 21.73 | 101.50 | 32.32 | 64.01 | 54.62 | 63.75 | 43.68 | 41.52 |

## 4.3 Ablations

We assess the impact of block size, number of diffusion steps, and choice of encoder-decoder architecture on downstream performance.

Table 7: Accuracy and decoding speed trade-off across varying block sizes for E2D2 on GSM8K. Decoding throughput (`Tput`) is measured in tokens / sec on 1 `A100 80GB` machine. We report mean $\pm$ standard deviation for 100 samples.

| Block size $S$ | 0-shot pass@1 ($\uparrow$) | Tput ($\uparrow$) |
|---|---|---|
| 32 | 20.9 | **62.2**$_{\pm 2.3}$ |
| 16 | 33.0 | 58.3$_{\pm 1.1}$ |
| 8 | 37.4 | 52.7$_{\pm 0.4}$ |
| 4 | 47.9 | 45.7$_{\pm 0.2}$ |
| 2 | **50.1** | 34.8$_{\pm 0.4}$ |

**Block size** For E2D2, using a larger block size increases throughput, as the encoder is invoked fewer times during generation. However, larger block sizes lead to worse quality, as the likelihood bound is less tight for larger blocks [1]. To explore this trade-off, we fine-tune E2D2 models with $N_{\text{Enc}}/N_{\text{Dec}} = 28/14$ and varying block sizes $S \in \{2, 4, 8, 16, 32\}$ on the GSM8K dataset. We report results in Table 7 where we observe the anticipated trade-off, with the best quality coming from the smallest block size $S = 2$ and the highest throughput coming from the largest block size $S = 32$.

**Number of Diffusion Steps** In Table 8, we show the effect of diffusion steps per block $T$ on sample quality using E2D2 and decoder-only BD3LM. For both models, we use $S = 4$. E2D2 outperforms BD3LM in quality for each $T$. As $T$ decreases, relative throughput gains from E2D2 diminish, since the decoder requires fewer invocations. For $T = 1$, E2D2 achieves comparable throughput to the 16-layer BD3LM because every sampling step invokes both the encoder and decoder, which together use comparable FLOPs relative to the 16-layer BD3LM. The 12-layer BD3LM achieves better throughput than E2D2 for $T = 1$, but sacrifices quality.

Table 8: Performance and decoding speed trade-off across varying number of diffusion steps $T$ for BD3LM and E2D2 on WMT. Decoding throughput (`Tput`) is measured in tokens / sec on 1 `A100 80GB` machine. Note that for all models, we use $L$ total sampling steps, so the throughput for diffusion models can in practice increase when $T < S$ for block size $S$. We report mean $\pm$ standard deviation for 100 samples. Best BLEU and throughput numbers are bolded.

|  | BD3LM
$N = 12$ | | BD3LM
$N = 16$ | | E2D2
$N_{\text{Enc}}/N_{\text{Dec}} = 28/4$ | |
|---|---|---|---|---|---|---|
| $T$ | BLEU ($\uparrow$) | Tput ($\uparrow$) | BLEU ($\uparrow$) | Tput ($\uparrow$) | BLEU ($\uparrow$) | Tput ($\uparrow$) |
| 1 | 17.2 | **149.2**$_{\pm 2.4}$ | 17.9 | 118.7$_{\pm 2.0}$ | 19.3 | 116.6$_{\pm 3.2}$ |
| 2 | 21.8 | 100.7$_{\pm 2.2}$ | 22.4 | 77.0$_{\pm 1.2}$ | 23.2 | 103.7$_{\pm 1.7}$ |
| 4 | 23.3 | 62.5$_{\pm 0.5}$ | 24.0 | 47.1$_{\pm 1.0}$ | **24.8** | 79.0$_{\pm 1.5}$ |

**Architecture Design** We compare the performance of the "last hidden state" and "shared KV cache" architecture variants (Section 3.2) when used to pretrain models for summarization and fine-tune models for math reasoning. In Table 9, we report perplexities from training models for 20k steps.

For models trained from scratch, e.g., on the larger summarization dataset, the "last hidden state" design performs best. We hypothesize that conditioning on the encoder's last hidden representation provides a richer signal than using intermediate representations. When fine-tuning from a strong Qwen3 model on the smaller GSM8K dataset, the "shared KV cache" variant achieves better perplexity. We speculate that by reusing the encoder's cached KVs, this design keeps the decoder's cross-attention inputs aligned with those expected by the base model, which facilitates more stable fine-tuning.

Table 9: Validation PPL upper bounds ($\downarrow$) for last hidden state and shared KV cache versions of E2D2 on CNN/DailyMail and GSM8K. † indicates model used in the main results. Best values are bolded.

|  | CNN/ DailyMail | GSM8K |
|---|---|---|
| Last Hidden State | **14.94**† | 1.98 |
| Shared KV Cache | 293.50 | **1.80**† |

## 5  Related Work

**Encoder-Decoder Models** T5 [54] and follow-up works FLAN and FLAN-UL2 [9, 64] demonstrated the efficacy of encoder-decoder architectures on a wide range of natural language benchmarks. Recently, this architecture was applied to the more updated Gemma family of models [65] in T5Gemma [79]. In contrast to our work, T5-style architectures use AR decoders and they separate out cross-attention between decoder and encoder representations into distinct modules, as opposed to performing attention over both decoder and encoder outputs in one forward pass, as in our work. Moreover, in these AR T5-style models, the encoder is only used once to provide an embedding of the input context. In E2D2, the encoder is applied to the context and each newly decoded block.

**Diffusion Language Models** Early approaches modeled word embeddings via Gaussian diffusion [24, 36]. Modern diffusion language models adopt a decoder-only architecture that is trained via denoising, similar to BERT [15]. The LLaDA model [46] scales masked diffusion models [45, 55] to the 8B parameter regime and demonstrated comparable and superior performance to a similarly-sized AR Llama model [17]. Another large discrete diffusion model Dream7B [78], which extends the DiffuLLaMA framework [22], uses pretrained AR models to initialize their models.

Unlike E2D2, these models only train with a full-sequence diffusion parameterization instead of the performant block diffusion framework [1]. When fine-tuning from AR checkpoints, this requires that models, such as Dream7B, carefully anneal the causal attention mask of the original pretrained AR LLM towards a fully bidirectional one. In contrast, for small block sizes $S \ll L$, our work does not require this tuned annealing as tokens only attend to tokens up to $S$ positions in the future.

**ENAT** Highly related to our work is the discrete image generation model ENAT [44], which leverages an encoder-decoder to decouple computational budget allocated to processing clean and masked tokens. The bidirectional nature of this non-AR model, however, precludes ENAT from leveraging efficient KV caching methods explored here. While our method supports variable-length sequence modeling, the ENAT architecture is only applicable to fixed-length outputs such as images.

**Efficient Inference** An analogy can be drawn between E2D2 and speculative decoding methods [8, 35], which speed up generation for AR models by utilizing smaller proposal networks. Most related to our work is Medusa [7], which uses several small MLPs to decode blocks of tokens conditioned on the final hidden state of the LLM. In contrast to our work however, these methods use accept/reject algorithms to keep proposal network tokens, while we use diffusion sampling.

## 6  Conclusion

In this work, we propose E2D2, a novel encoder-decoder architecture with efficient diffusion training and sampling algorithms. Our results demonstrate that E2D2 better trades off throughput and sample quality relative to prior state-of-the-art masked diffusion models. While E2D2 represents a promising direction to narrow both the performance and throughput gaps relative to AR models, further innovation is required to increase training efficiency and sample quality. Finally, given we are working on language modeling, we carry the inherent risks and opportunities of this line of research.

## Acknowledgments

This work was partially funded by the National Science Foundation under award CAREER 2145577, and by the National Institute of Health under award MIRA R35GM151243. Marianne Arriola is supported by a NSF Graduate Research Fellowship under award DGE-2139899 and a Hopper-Dean/Bowers CIS Deans Excellence Fellowship. We gratefully acknowledge use of the research computing resources of the Empire AI Consortium, Inc, with support from Empire State Development of the State of New York, the Simons Foundation, and the Secunda Family Foundation [5]. This research was also supported in part through the use of computational resources provided by Lambda (`lambda.ai`) in partnership with Open Athena AI Foundation, Inc. We gratefully acknowledge their generous GPU infrastructure grants that helped make this work possible and thank Eric Czech of Open Athena for the useful discussions and feedback.

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

# Contents

# A   Extended Background on Block Diffusion Language Models

Below, we provide an extended overview of block diffusion language models closely following [1]. In particular, we define the masking process $q(\mathbf{z}_t^{1:B} \mid \mathbf{z}^{1:B})$ and the derivation of the objective $\mathcal{L}(\mathbf{x}^{1:B}; \theta)$ for parameters $\theta$.

Recall that we factorize the sequence $\mathbf{x}^{1:L}$ drawn from the data distribution $q(\mathbf{x}^{1:L})$ over $B$ blocks of size $S$. We simplify notation by denoting the tokens $\mathbf{x}^{(b-1)\cdot S : b \cdot S}$ in a block $b$ as $\mathbf{x}^b$. We perform diffusion in each block over $T$ discretization steps. We define $t, s$ to denote $t(i) = i/T$ and $s(i) = (i-1)/T$, for all $i \in [1, T]$. We denote the Kullback-Leibler divergence as $D_{\mathrm{KL}}[\cdot]$. Below we reproduce the negative evidence lower bound (NELBO) following [1]:

$$
\begin{aligned}
-\log p_\theta(\mathbf{x}) &= -\sum_{b=1}^{B} \log p_\theta(\mathbf{x}^b | \mathbf{x}^{<b}) \\
&= -\sum_{b=1}^{B} \log \mathbb{E}_q \frac{p_\theta(\mathbf{z}_{t(1):t(T)}^b | \mathbf{x}^{<b})}{q(\mathbf{z}_{t(1):t(T)}^b | \mathbf{x}^b)} \\
&= -\sum_{b=1}^{B} \log \mathbb{E}_q \frac{p_\theta(\mathbf{z}_{t(T)}^b | \mathbf{x}^{<b}) \prod_{i=1}^{T} p_\theta(\mathbf{z}_{s(i)}^b | \mathbf{z}_{t(i)}^b, \mathbf{x}^{<b})}{\prod_{i=1}^{T} q(\mathbf{z}_{t(i)}^b | \mathbf{z}_{s(i)}^b)} \\
&\leq \sum_{b=1}^{B} \Bigg[ \underbrace{-\mathbb{E}_q \log p_\theta(\mathbf{x}^b | \mathbf{z}_{t=\frac{1}{T}}^b, \mathbf{x}^{<b})}_{\mathcal{L}_{\text{recons}}} \\
&\quad + \underbrace{\mathbb{E}_{t \in \left\{ \frac{2}{T}, \dots, \frac{T-1}{T}, 1 \right\}} \mathbb{E}_q T D_{KL} \left( q(\mathbf{z}_s^b | \mathbf{z}_t^b, \mathbf{x}^b) \parallel p_\theta(\mathbf{z}_s^b | \mathbf{z}_t^b, \mathbf{x}^{<b}) \right)}_{\mathcal{L}_{\text{diffusion}}} \\
&\quad + \underbrace{D_{KL} \left( q(\mathbf{z}_{t=1}^b | \mathbf{x}^b) \parallel p_\theta(\mathbf{z}_{t=1}^b) \right)}_{\mathcal{L}_{\text{prior}}} \Bigg]
\end{aligned}
\tag{1}
$$

## A.1   Forward Noise Process

Within each block, [1] adopt the masked diffusion language modeling framework, which is the most performant [4]. They use a simplified NELBO as proposed by [47, 55, 58].

Following [4], we first define a diffusion matrix $Q_t \in \mathbb{R}^{V \times V}$ corresponding to noise level $t$ for states $i \in \{1, \dots, V\}$ and vocabulary size $V$. Recall that the prior distribution is the absorbing mask state $\mathbf{m}$, a one-hot vector centered on the special [MASK] token index. We denote this mask index as the last token in the vocabulary, $\arg\max_i \mathbf{m}_i = V$. Consider the noise schedule function $\alpha_t \in [0, 1]$, which is monotonically decreasing in $t$, with $\alpha_0 = 1$ and $\alpha_1 = 0$. Then, the diffusion matrix is:

$$
[Q_t]_{ij} = \begin{cases} 1 & \text{if } i = j = V \\ \alpha_t & \text{if } i = j \neq V \\ 1 - \alpha_t & \text{if } j = V, i \neq V \end{cases}
\tag{2}
$$

For the forward marginal $Q_{t|s}$, we use forward marginal probabilities according to $\alpha_{t|s} = \alpha_t / \alpha_s$. The forward noise process is applied independently for each token $\ell \in \{1, \dots L\}$. We first define the transition matrix $\overline{Q}_{t(i)} = Q_{t(1)} Q_{t(2)} \dots Q_{t(i)}$ which is used in the forward process $q(\mathbf{z}_t^\ell | \mathbf{x}^\ell) = \mathrm{Cat}\left(\mathbf{z}_t^\ell; \overline{Q}_t \mathbf{x}^\ell\right)$.

## A.2 Reverse Denoising Process

We now may obtain the reverse posterior $q(\mathbf{z}_s^\ell \mid \mathbf{z}_t^\ell, \mathbf{x}^\ell)$ derived in D3PM [4] as follows, where $\odot$ denotes the Hadamard product between two vectors:

$$q(\mathbf{z}_s^\ell|\mathbf{z}_t^\ell,\mathbf{x}^\ell) = \frac{q(\mathbf{z}_t^\ell|\mathbf{z}_s^\ell,\mathbf{x}^\ell)q(\mathbf{z}_s^\ell|\mathbf{x}^\ell)}{q(\mathbf{z}_t^\ell|\mathbf{x}^\ell)} = \mathrm{Cat}\left(\mathbf{z}_s^\ell; \frac{Q_{t|s}\mathbf{z}_t^\ell \odot Q_s^\top \mathbf{x}^\ell}{(\mathbf{z}_t^\ell)^\top Q_t^\top \mathbf{x}^\ell}\right) \tag{3}$$

## A.3 Negative Evidence Lower Bound (NELBO)

As in block diffusion [1], we adopt the simplified objective for masked diffusion [47, 55, 58] to obtain a tighter NELBO. We provide the sketch for the derivation, with the full proof is provided in [47, 55, 58].

We first focus on simplifying the diffusion loss term in (1). To simplify the denoising model, we enforce the following constraints on the design of the denoising network by taking advantage of the fact that there only exists two possible states in the diffusion process $\mathbf{z}_t^\ell \in \{\mathbf{x}^\ell, \mathbf{m}\} \ \forall \ell \in \{1, \dots, L\}$ [55].

1. **Zero Masking Probabilities**. The clean sequence does not contain masks. So, we set $p_\theta(\mathbf{x}^\ell = \mathbf{m} \mid \mathbf{z}_t^\ell) = 0$.
2. **Carry-Over Unmasking**. If a token is unmasked in the reverse denoising process, it is never remasked by definition of the masked diffusion process. The true reverse posterior for the case where $\mathbf{z}_t^\ell \neq \mathbf{m}$ is $q(\mathbf{z}_s^\ell = \mathbf{z}_t^\ell|\mathbf{z}_t^\ell \neq \mathbf{m}) = 1$. Thus, we set $p_\theta(\mathbf{z}_s^\ell = \mathbf{z}_t^\ell|\mathbf{z}_t^\ell \neq \mathbf{m}) = 1$.

With these two simplifications, we now use our denoising model to parameterize the posterior $p_\theta(\mathbf{z}_s^\ell = \mathbf{x}^\ell|\mathbf{z}_t^\ell = \mathbf{m})$. Let $\mathbf{x}^{b,\ell}$ denote a token in the $\ell$-th position in block $b \in \{1, \dots, B\}$. The diffusion loss term is derived as follows:

$$\mathcal{L}_{\mathrm{diffusion}} = \sum_{b=1}^{B} \mathbb{E}_t \mathbb{E}_q T \left[ D_{\mathrm{KL}} \left[ q(\mathbf{z}_s^b|\mathbf{z}_t^b, \mathbf{x}^b) \| p_\theta(\mathbf{z}_s^b|\mathbf{z}_t^b, \mathbf{x}^{<b}) \right] \right]$$

$$= \sum_{b=1}^{B} \mathbb{E}_t \mathbb{E}_q T \left[ \sum_{\ell=1}^{L'} D_{\mathrm{KL}} \left[ q(\mathbf{z}_s^{b,\ell}|\mathbf{z}_t^{b,\ell}, \mathbf{x}^{b,\ell}) \| p_\theta(\mathbf{z}_s^{b,\ell}|\mathbf{z}_t^b, \mathbf{x}^{<b}) \right] \right]$$

$D_{\mathrm{KL}}$ is simply the discrete-time diffusion loss for the block $b$; hence, from [55] (Suppl. B.1), we get:

$$= \sum_{b=1}^{B} \mathbb{E}_t \mathbb{E}_q T \left[ \sum_{\ell=1}^{L'} \frac{\alpha_t - \alpha_s}{1 - \alpha_t} \log p_\theta(\mathbf{x}^{b,\ell} \mid \mathbf{z}_t^{b,\ell}, \mathbf{x}^{<b}) \right]$$

$$= \sum_{b=1}^{B} \mathbb{E}_t \mathbb{E}_q T \left[ \frac{\alpha_t - \alpha_s}{1 - \alpha_t} \log p_\theta(\mathbf{x}^b \mid \mathbf{z}_t^b, \mathbf{x}^{<b}) \right] \tag{4}$$

Lastly, we obtain a tighter approximation of the likelihood by taking the diffusion steps $T \to \infty$ [55], for which $T(\alpha_t - \alpha_s) = \alpha_t'$:

$$\mathcal{L}_{\mathrm{diffusion}} = \sum_{b=1}^{B} \mathbb{E}_{t \sim [0,1]} \mathbb{E}_q \left[ \frac{\alpha_t'}{1 - \alpha_t} \log p_\theta(\mathbf{x}^b \mid \mathbf{z}_t^b, \mathbf{x}^{<b}) \right] \tag{5}$$

When taking $T \to \infty$, [55] (Suppl. A.2.4) show the reconstruction loss becomes 0. In particular, we use the fact that $\mathbf{z}_{t(1)}^b \sim \lim_{T \to \infty} \mathrm{Cat}\left(.; \mathbf{z}_{t=\frac{1}{T}}^b\right) = \mathrm{Cat}(.; \mathbf{x}^b)$. Then, we obtain:

$$\mathcal{L}_{\mathrm{recons}} = -\mathbb{E}_q \log p_\theta(\mathbf{x}^b|\mathbf{z}_{t(1)}^b, \mathbf{x}^{<b})$$

$$= -\log p_\theta(\mathbf{x}^b|\mathbf{z}_{t(1)}^b = \mathbf{x}^b, \mathbf{x}^{<b})$$

$$= 0 \tag{6}$$

Also, the prior loss $\mathcal{L}_{\mathrm{prior}} = D_{\mathrm{KL}}\left(q(\mathbf{z}_{t=1}^b|\mathbf{x}^b) \| p_\theta(\mathbf{z}_{t=1}^b)\right)$ reduces to 0 since $\alpha_{t=1} = 0$ so that $q(\mathbf{z}_{t=1}^b|\mathbf{x}^b) = \mathrm{Cat}(.; \mathbf{m})$ and $p_\theta(\mathbf{z}_{t=1}^b) = \mathrm{Cat}(.; \mathbf{m})$; see [55] (Suppl. A.2.4).

The final diffusion objective simply becomes a weighted average of cross-entropy terms:

$$\mathcal{L}_{\text{BD}}(\mathbf{x}; \theta) = \sum_{b=1}^{B} \mathbb{E}_{t \sim [0,1]} \mathbb{E}_q \left[ \frac{\alpha'_t}{1 - \alpha_t} \log p_\theta(\mathbf{x}^b \mid \mathbf{z}^b_t, \mathbf{x}^{<b}) \right] \tag{7}$$

## B  Training FLOPs derivation

We derive the training FLOPs of a model forward pass in Table 1 as follows. Let $L$ be the sequence length, $N$ be the number of layers, and $D$ be the hidden dimension. We calculate the attention FLOPs of the forward pass following FlashAttention [12], which is $4NDL^2$ for bidirectional attention, as each matrix multiplication (i.e., multiplying queries and keys and multiplying the attention output by values) requires $2NDL^2$ FLOPs. Assuming a gated MLP with three dense layers [13] and intermediate hidden dimension of $4D$, the MLP FLOPs are given as $24NLD^2 = 2NL(D)(4D)$ (up-projection) $+ 2NL(D)(4D)$ (gate projection) $+ 2NL(4D)(D)$ (down-projection).

**Autoregressive.**   The number of attention operations in causal attention is: $\sum_{i=1}^{L} i = \frac{L^2+L}{2}$. Thus, AR training uses $4ND(\frac{L^2+L}{2})$ attention FLOPs. The MLP FLOPs are given as $24NLD^2$.

**Full-Sequence Diffusion.**   Sequence diffusion models such as Masked Diffusion LMs [47, 55, 58] training uses $4NDL^2$ attention FLOPs from bidirectional attention, requiring $L^2$ attention operations. The MLP FLOPs are also $24NLD^2$.

**Block Diffusion.**   For block diffusion models [1], the sequence is split into $B$ blocks of size $S$, where $L = BS$.

The training algorithm proposed in BD3LM [1] is $2\times$ more computationally expensive relative to both AR and full-sequence diffusion. Below, we summarize the attention operations used to compute the attention FLOPs from [1].

BD3LM uses attention mask $\mathcal{M}_{\text{BD3LM}} \in \{0,1\}^{2L \times 2L}$ for both the noised tokens $\mathbf{z}^{1:L}_t$ and clean tokens $\mathbf{x}^{1:L}$. This mask can visualized as a concatenation of the encoder and decoder masks from Figure 2. This mask is thus comprised of four $L \times L$ smaller attention masks:

$$\mathcal{M}_{\text{BD3LM}} = \begin{bmatrix} \mathbf{0} & \mathcal{M}_{\text{BC}} \\ \mathcal{M}_{\text{OBC}} & \mathcal{M}_{\text{BD}} \end{bmatrix}$$

where $\mathcal{M}_{\text{BD}}$ and $\mathcal{M}_{\text{OBC}}$ are used to update the representation of $\mathbf{z}^{1:L}_t$ and $\mathcal{M}_{\text{BC}}$ is used to update the representation of $\mathbf{x}^{1:L}$. We define these masks as follows:

1. Block-causal self-attention mask to update clean $\mathbf{x}^{1:B}$. $\mathcal{M}_{\text{BC}} \in \{0,1\}^{L \times L}$

$$[\mathcal{M}_{BC}]_{ij} = \begin{cases} 1 & \text{if } j \text{ belongs in the same block as } i, \text{ or a block before } i \\ 0 & \text{otherwise} \end{cases}$$

   Thus, $\mathcal{M}_{\text{BC}}$ is nonzero for $\sum_{b=1}^{B} bS^2 = \frac{S^2 B^2 + BS^2}{2} = \frac{L^2 + LS}{2}$ entries

2. Block-diagonal self-attention mask to update noised $\mathbf{z}^{1:B}_t$. $\mathcal{M}_{\text{BD}} \in \{0,1\}^{L \times L}$

$$[\mathcal{M}_{BD}]_{ij} = \begin{cases} 1 & \text{if } i, j \text{ are in the same block} \\ 0 & \text{otherwise} \end{cases}$$

   Thus, $\mathcal{M}_{\text{BD}}$ is nonzero for $BS^2 = LS$ entries.

3. Offset block-causal cross-attention mask to update $\mathbf{z}^{1:B}_t$ via cross-attention to $\mathbf{x}^{1:B}$

$$[\mathcal{M}_{OBC}]_{ij} = \begin{cases} 1 & \text{if } j \text{ belongs in a block before } i \\ 0 & \text{otherwise} \end{cases}$$

   Thus, $\mathcal{M}_{\text{OBC}} = \mathcal{M}_{\text{BC}} - \mathcal{M}_{\text{BD}}$ such that the number of nonzero entries is $\frac{L^2 + LS}{2} - LS = \frac{L^2 - LS}{2}$.

Thus, the total number of attention operations is $\frac{L^2+LS}{2} + LS + \frac{L^2-LS}{2} = L^2 + LS$.

The number of MLP FLOPs is $48NLD^2$ as the input sequence to each layer is of length $2L$ from both $\mathbf{x}^{1:L}$ and $\mathbf{z}_t^{1:L}$.

**Encoder-Decoder Block Diffusion (Ours).** E2D2 uses half the attention FLOPs compared to BD3LM by splitting the operation into encoder and decoder layers. This is because each encoder and decoder layer uses the same number of attention FLOPs.

1. Encoder attention operations. The encoder uses mask $\mathcal{M}_{\text{Enc}} \in \{0,1\}^{L\times L} = \mathcal{M}_{\text{BC}}$. Thus, the encoder uses $\frac{L^2+LS}{2}$ attention operations

2. Decoder attention operations. The decoder uses mask $\mathcal{M}_{\text{Dec}} \in \{0,1\}^{L\times 2L} = \mathcal{M}_{\text{OBC}}\oplus\mathcal{M}_{\text{BD}}$. Thus, the decoder uses $\frac{L^2-LS}{2} + LS = \frac{L^2+LS}{2}$ attention operations.

The FLOPs of an attention pass is $4(N_{\text{Enc}} + N_{\text{Dec}})D\frac{L^2+LS}{2}$. The MLP FLOPs is $24NLD^2$ since the input sequence contains $L$ tokens at each layer.

## C    FLOPs comparison to MDLM

Below, we show that E2D2 attention FLOPs for the block diffusion parameterization are upper-bounded by decoder-only MDLM FLOPs under the same model size. Assume both models have the same number of total layers, where $N = N_{\text{Enc}} + N_{\text{Dec}}$. Then,

$$\text{FLOPs}_{\text{E2D2}} \leq \text{FLOPs}_{\text{MDLM}} \tag{8}$$
$$4ND\left(\frac{L^2+LS}{2}\right) \leq 4NDL^2$$
$$\left(\frac{L^2+LS}{2}\right) \leq L^2$$
$$\left(\frac{L+S}{2}\right) \leq L$$
$$S \leq L \tag{9}$$

So, as long as $S < L$, E2D2 FLOPs are fewer than that of MDLM under equal model size. When $S = L$, we exactly recover MDLM FLOPs.

## D    MDLM with Encoder-Decoder Architecture

**Notation**    Recall that $\mathbf{x}^{1:P}$ denotes $P$ prompt tokens. We denote the set of unmasked token indices in $\mathbf{z}_t^{1:L}$ as $C_t = \{\ell \mid \mathbf{z}_t^\ell \neq \mathbf{m}\}$, and we define $\mathbf{x}^{C_t} = \bigoplus_{\ell \in C_t} \mathbf{z}_t^\ell$. The encoder receives inputs $\mathbf{x}_{t,\text{Enc}} = \mathbf{x}^{1:P} \oplus \mathbf{x}^{C_t}$ and produces representations $\mathbf{h}_t$.

**Sampling**    In Algorithm 3, we present the sampling procedure for MDLM parameterized by an encoder-decoder architecture. We denote the denoising steps budget as $T$, and the reverse diffusion process proceeds as $t = 1, \frac{T-1}{T} \ldots, \frac{1}{T}$. We also denote the number of steps where only the decoder is called by $n$. Thus, the encoder will be invoked at time steps $t = 1, \frac{T-n}{T}, \frac{T-2n}{T}, \ldots$.

We begin by encoding the $P$ prompt tokens $\mathbf{x}^{1:P}$ to attain the encoder representation $\mathbf{h}^{1:P}$. The decoder's input is initialized to a sequence of length $L$ consisting of only masked tokens. The decoder iteratively denoises this full sequence for a fixed number of steps $n$ with cross-attention to $\mathbf{h}_1 = \mathbf{h}^{1:P}$. Later in the decoding process, in some interval $[s, t]$ (with $s = t - \frac{n}{T}$), during which only the decoder is invoked over $n$ evaluations and the encoder is not invoked, we treat $\mathbf{h}_\tau$ as constant for all $\tau \in [s,t]$. At $s$, the decoder returns any new tokens decoded from the interval $[s,t]$, which are incorporated into the encoder's input, and the encoder is evaluated to produce an updated $\mathbf{h}_s$. This process is repeated until the sampling budget $T$ is exhausted.

---

**Algorithm 3** Encoder-Decoder MDLM Sampling

---

**Require:** Prompt $\mathbf{x}^{1:P}$, number of steps during which only the decoder is invoked $n$, number diffusion steps $T$,
    ENCODER, DECODER, algorithm SAMPLE
1: $\mathbf{h} \leftarrow \text{ENCODER}(\mathbf{x}^{1:P})$                                                    ▷ Encode prompt
2: $\mathbf{z}_t^{1:L} \sim \mathbf{m}^{1:L}$                                        ▷ Initialize fully noised sequence
3: **for** $i = T, T - n, T - 2n, \ldots, n$ **do**
4:     $t \leftarrow \frac{i}{T}$
5:     **for** $n$ steps **do**
6:         $\mathbf{z}_t^{1:L} \leftarrow \text{SAMPLE}(\text{DECODER}(\mathbf{z}_t^{1:L}, \mathbf{h}))$          ▷ Denoise sequence: Apply small decoder only
7:     **end for**
8:     $\mathbf{x}^{C_t} \leftarrow$ Unmasked tokens in $\mathbf{z}_t^{1:L}$                      ▷ Accumulate output
9:     $\mathbf{x}_{t,\text{Enc}} \leftarrow \mathbf{x}^{1:P} \oplus \mathbf{x}^{C_t}$
10:    $\mathbf{h} \leftarrow \text{ENCODER}(\mathbf{x}_{t,\text{Enc}})$                               ▷ Encode new 'block'
11: **end for**
12: **return** $\mathbf{z}_t^{1:L}$

---

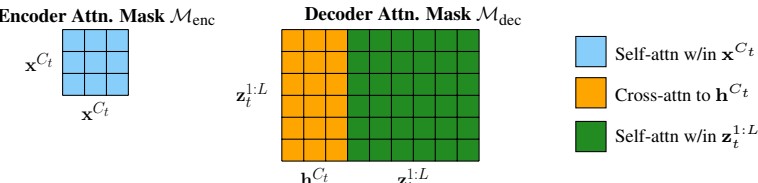

Figure 4: Example attention masks used in training full-sequence diffusion with an encoder-decoder architecture, for $L = 6$ tokens. We denote clean tokens in $\mathbf{z}_t^{1:L}$ as $\mathbf{x}^{C_t}$, where $C_t$ corresponds to their token indices. In this example, $|C_t| = 3$. **Left:** The encoder mask $\mathcal{M}_{\text{Enc}} \in \{0,1\}^{|C_t| \times |C_t|}$ enables clean tokens to attend to other clean tokens in the sequence. **Right:** The decoder mask $\mathcal{M}_{\text{Dec}} \in \{0,1\}^{L \times (L+|C_t|)}$ uses bidirectional attention across the noised sequence $\mathbf{z}_t^{1:L}$ and cross-attention to the encoder's output representation $\mathbf{h}^{C_t}$.

**Training** During training, we sample $\mathbf{z}_t^{1:L} \sim q(\mathbf{z}_t^{1:L} \mid \mathbf{x}^{1:L})$. The input to the encoder is the sequence of unmasked tokens $\mathbf{x}^{C_t}$ from the full latent sequence $\mathbf{z}_t^{1:L}$ (along with the prompt tokens $\mathbf{x}^{1:P}$, if available), and it produces a representation of these tokens $\mathbf{h}_t$. The input to the decoder is $\mathbf{z}_t^{1:L}$ along with $\mathbf{h}_t$, and it outputs logits for each token position in $\mathbf{z}_t^{1:L}$.

In Figure 4, we present a sample $\mathcal{M}_{\text{Enc}}$ and $\mathcal{M}_{\text{Dec}}$ for training MDLM with E2D2. Both the encoder and the decoder use full bidirectional self- and cross-attention. For batched training, where the number of unmasked tokens can differ between sequences, we may pad the encoder input sequences: $\mathbf{x}^{C_t} \oplus (\bigoplus_{\ell \notin C_t} [\text{PAD}])$ where [PAD] is a special padding token. We also adjust the attention masks $\mathcal{M}_{\text{Enc}}$ and $\mathcal{M}_{\text{Dec}}$ accordingly, to disable attention to padding tokens.

# E    Additional Experimental Details

For all of our experiments, we used architectures based on the Qwen3 family of models [76]. In all training and fine-tuning experiments, we used the ADAM optimizer [31] with weight decay $1e^{-5}$ and $(\beta_1, \beta_2) = (0.9, 0.98)$. We also apply gradient clipping to $1.0$. We also utilize an exponential moving average (EMA) copy of each model with a decay rate of $0.9999$. The EMA model was used during evaluation.

## E.1    Measuring Throughput

For the main results, inference throughput was measured on a single H100 (80GB) GPU using a batch size of 1. For the ablation results, we used a single A100 (80GB) GPU. We first processed 100 'warm-up' samples and then recorded decoding throughput for the subsequent 100 samples. Models generated 256 additional tokens, with no [EOS] or other early decoding stopping criteria applied. We report mean $\pm$ the standard deviation of tokens per second for the 100 generated samples.

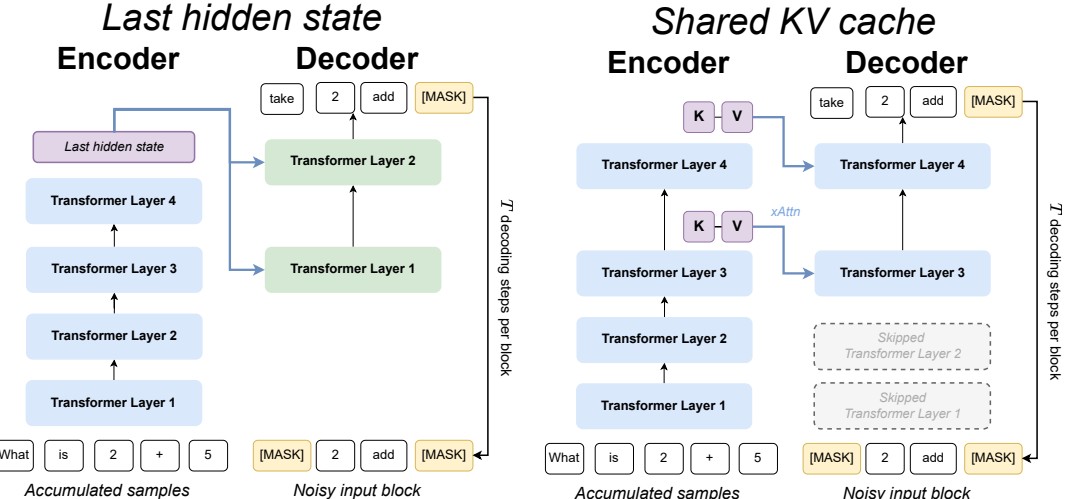

Figure 5: Encoder-decoder architecture for discrete diffusion. The encoder embeds the clean sequence. The lightweight decoder denoises blocks over $T$ decoding steps by cross-attending to the encoder output. **Left:** "Last hidden state" version of E2D2: each decoder block attends to the last hidden state output of the encoder. **Right:** "Shared KV cache" version of E2D2: Decoder layers attend to the keys and values of corresponding encoder layers. Encoder and decoder weights are tied.

## E.2   Summarization

**Data**   For this task, we use the CNN/DailyMail dataset version 3.0 [27, 57] downloaded from https://huggingface.co/datasets/abisee/cnn_dailymail. Data was pre-processed to add a prefix to summarizations: "Summary:   ". Inputs and targets were truncated to a maximum length of 512 each, ensuring a maximum sequence length of 1024 for sequences seen during training.

**Hyperparameters**   We used the Qwen/Qwen3-0.6B-Base tokenizer. All models had hidden size of 256 and intermediate hidden size of 768. The AR and MDLM baselines consisted of 28 transformer layers, corresponding to 80M parameters. BD3LM had 12 layers and used a block size of $S = 8$, corresponding to 60M parameters. E2D2 also used $S = 8$ and consisted of 20 encoder and 8 decoder layers, corresponding to 80M parameters. We used the "last hidden state variant" of the E2D2 model.

We trained with batch size 128. Learning rate was linearly warmed-up for 1000 steps until a maximum of $3\mathrm{e}^{-4}$. Models were trained for a maximum of 500k steps and we use early stopping on the validation loss to select the best model.

**Evaluation**   Models generated 256 additional tokens. Metrics were computed using the evaluate library from HuggingFace. Inputs were pre-processed as above, with the exception that input texts were truncated to 2048 tokens, and generated samples were post-processed to truncate summaries after the end of sentence special tokens. For MDLM, BD3LM, and E2D2, we apply exponential length penalty starting at 80 tokens with a factor of 1.1 and repetition penalty with a factor of 1.5. For the AR model, we do not apply the length or repetition penalties as these led to a performance degradation.

For the BD3LM model, we found improved performance when aligning the input sequence to block length. That is, for a context of length $|C|$ the last $C \mod S$ tokens were fixed to the first decoding block. For E2D2 and MDLM models, this alignment was not necessary. Finally, for MDLM we decoded semi-autoregressively using a block size of 32.

## E.3   Translation

**Data**   For this task, we use the WMT14 German to English (de-en) dataset [6] downloaded from https://huggingface.co/datasets/wmt/wmt14. For AR models, data was pre-processed to

add a prefix to the translations: "`Translation: `". Inputs and targets were truncated to a maximum length of 128 each, ensuring a maximum sequence length of 256 for sequences seen during training.

**Hyperparameters**   We used the `Qwen/Qwen3-0.6B-Base` tokenizer. All models had hidden size of 512 and intermediate hidden size of 1536. The AR and MDLM baselines consisted of 32 transformer layers, corresponding to 250M parameters. We trained two BD3LM models, one with 12 layers and one with 16 and used a block size of $S = 4$. This corresponds to 140M and 160M parameters, respectively. E2D2 also used $S = 4$ and consisted of 28 encoder and 4 decoder layers, corresponding to 250M parameters. We used the "last hidden state variant" of the E2D2 model.

We trained with batch size 128. Learning rate was linearly warmed-up for 1000 steps until a maximum of $3e^{-4}$. Models were trained for a maximum of 500k steps and we use early stopping on the validation loss to select the best model.

**Evaluation**   Models generated 256 additional tokens. Metrics were computed using the `evaluate` library from HuggingFace. Inputs were pre-processed as above and generated samples were post-processed to truncate translations after the first '.' or the end of sentence special token.

For the BD3LM model, we found improved performance when aligning the input sequence to block length. For E2D2 and MDLM models, this alignment was not necessary. Finally, for MDLM we decoded semi-autoregressively using a block size of 32.

### E.4   Mathematical Reasoning

**Data**   For this task, we use the GSM8K dataset [10] downloaded from `https://huggingface.co/datasets/openai/gsm8k` (`main` version). Data was pre-processed to add a prefix to inputs: "`Please reason step by step, and put your final answer within $\boxed{}$.`", answers were prefaced with "`Answer: `", and solution was wrapped in "`$\boxed{}$.`". Inputs and targets were truncated to a maximum length of 384 each, ensuring a maximum sequence length of 768 for sequences seen during training.

**Hyperparameters**   We used the `Qwen/Qwen3-1.7B-Base` tokenizer. For AR and MDLM models, we used all the layers of the pretrained 1.7B parameter Qwen3, which has a hidden size of 2048 and intermediate hidden size of 6144. For the BD3LM baselines, we used the $N$ layers closest to the input, as opposed to the $N$ layers closest to the output, since we found this to be the stronger of the two baselines. For our BD3LM baseline where $N = 21$, this corresponds to 1.4B parameters. For the E2D2 decoder, we used the $N_{\mathrm{Dec}}$ layers closest to the output. We used the "shared KV cache variant" of the E2D2 model and the encoder and decoder layers were weight-tied. Thus, E2D2 uses 1.7B trainable parameters. BD3LM and E2D2 models were trained using block size of $S = 4$.

We trained with batch size 1. Learning rate was linearly warmed-up for 100 steps until a maximum of $1e^{-5}$ and decayed using a cosine schedule until half the maximum value. Models were trained for a maximum of 30k steps and we use early stopping on the validation loss to select the best model.

**Evaluation**   Models generated 512 additional tokens. Inference was done using the `lm-eval harness` library with the 'flexible match' criteria and similar pre-processing was applied to question texts. Post-processing was done to truncate text at end of sentence special characters and solutions were reverted to their original form of `### <Answer>`. Inputs were pre-processed as above.

For both the BD3LM and E2D2 models, we found improved performance when aligning the input sequence to block length.

### E.5   Language Modeling

**Data**   We train on the OpenWebText [21] dataset downloaded from `https://huggingface.co/datasets/Skylion007/openwebtext`. We use the GPT2 tokenizer. We do not pad or truncate sequences, but concatenate them and wrap them to a length of 1024. Since OWT does not have a validation split, we leave the last 100k documents for validation. Following [1], we do not inject [BOS] and [EOS] tokens at the beginning and end of each sequence in order to enable arbitrary-length generation.

We evaluate zero-shot likelihoods on datasets Penn Tree Bank (PTB; [41]), Wikitext [42], LM1B, Lambada [49], AG News [80], and Scientific Papers (Pubmed and Arxiv subsets; [11]). When reporting zero-shot likelihoods on benchmark datasets from [53], we wrap all sequences to 1024 tokens and do not add [EOS] between sequences following [55].

**Hyperparameters**  We train with a batch size of 512 and context length of 1024 for 1M steps. We trained E2D2 using a block size of $S = 4$. All models use $N = 12$ total layers, and for E2D2 we set $N_{\text{Enc}} = 10, N_{\text{Dec}} = 2$. All models use 170M total parameters, including token embeddings. For E2D2, the learning rate was linearly warmed-up for 2000 steps until a constant peak of $3e^{-4}$.

**Evaluation**  We report validation likelihood for OWT, and we use the test splits for reporting zero-shot likelihoods using the datasets from [53]. We use the same context length of 1024 tokens at evaluation.

## F  Assets

In Table 10, we list the corresponding licenses for datasets used in this work.

Table 10: Datasets and corresponding licenses.

| Dataset | Licence |
|---|---|
| CNN/DailyMail | Apache-2.0 License |
| WMT14 | — |
| GSM8K | MIT License |
| OpenWebText [21] | Creative Commons CC0 license ("no rights reserved") |

In Table 11, we list the corresponding licenses for software packages used in this work.

Table 11: Software and corresponding licenses.

| Library | License |
|---|---|
| HuggingFace [71] | Apache 2.0 |
| Hydra [75] | MIT |
| Language Model Evaluation Harness [18] | MIT |
| Matplotlib [29] | Matplotlib license |
| MosaicML Composer [66] | Apache 2.0 |
| NumPy [25] | NumPy license |
| OmegaConf | BSD 3-Clause |
| Pandas [48] | BSD 3-Clause "New" or "Revised" |
| PyTorch [51] | BSD-3 Clause |
| Seaborn [70] | BSD 3-Clause "New" or "Revised" |
| TorchMetrics | Apache 2.0 |

## G  Generated Samples

Below, we provide example generations from GSM8K from E2D2 $N_{\text{Enc}}/N_{\text{Dec}} = 28/14$ and BD3LM $N = 21$.

### G.1  E2D2

---

Question:  Every day, Wendi feeds each of her chickens three cups of mixed
chicken feed, containing seeds, mealworms and vegetables to help keep them
healthy.  She gives the chickens their feed in three separate meals.  In
the morning, she gives her flock of chickens 15 cups of feed.  In the
afternoon, she gives her chickens another 25 cups of feed.  How many cups

of feed does she need to give her chickens in the final meal of the day if the size of Wendi's flock is 20 chickens?

Answer:  Wendi feeds her chickens 3 cups of feed per day for 20 chickens for a total of 3*20 = «3*20=60»60 cups of feed.  In the morning, she gives her chickens 15 cups of feed and in the afternoon, she gives them another 25 cups of feed for a total of 15+25 = «15+25=40»40 cups of feed.  She needs 60 cups of feed and has already given 40 cups so she needs to give her chickens 60-40 = «60-40=20»20 cups of feed in the final meal #### 20

---

Question:  Josh decides to try flipping a house.  He buys a house for $80,000 and then puts in $50,000 in repairs.  This increased the value of the house by 150%.  How much profit did he make?

---

Answer:  The repairs cost 50,000*.5=$«50000*.5=25000»25,000.  So the house was worth for 80,000+25,000=$«80000+25000=105000»105,000.  So he made a profit of 105,000-80,000=$«105000-80000=25000»25,000 #### 25000

## G.2  BD3LM ($N = 21$)

---

Question:  Every day, Wendi feeds each of her chickens three cups of mixed chicken feed, containing seeds, mealworms and vegetables to help keep them healthy.  She gives the chickens their feed in three separate meals.  In the morning, she gives her flock of chickens 15 cups of feed.  In the afternoon, she gives her chickens another 25 cups of feed.  How many cups of feed does she need to give her chickens in the final meal of the day if the size of Wendi's flock is 20 chickens?

Answer:  In the morning, Wendi gives her chickens 15 cups of feed.  In the afternoon, she gives her chickens another 25 cups of feed.  In the final, she gives her chickens 20 * 3 = «20*3=60»60 cups of feed.  Thus, in the final meal of the day, Wendi gives her chickens 60 - 25 - 15 = «60-25-15=10»10 cups of feed.  #### 10

---

Question:  Josh decides to try flipping a house.  He buys a house for $80,000 and then puts in $50,000 in repairs.  This increased the value of the house by 150%.  How much profit did he make?

Answer:  The value of the house increased by 50000*.15=$«50000*.15=7500»7500
So the value of the house was 80000+7500=$«80000+7500=87500»87500
So he made 87500-80000=$«87500-80000=7500»7500 #### 7500

---

