# OpenReview forum: "Encoder-Decoder Diffusion Language Models for Efficient Training and Inference"
_NeurIPS.cc/2025/Conference — NeurIPS 2025 poster_

### Official Review · Reviewer_xEvo · 2025-06-21

**Clarity:** 3
**Significance:** 2
**Originality:** 2
**Rating:** 3
**Confidence:** 2

**Summary:**

This paper introduces E2D2, a novel language model that generates text by sequentially creating "blocks" and filling each block in parallel using a diffusion process. This hybrid approach allows the model to achieve faster inference speeds than traditional methods while striking a better balance between text quality and computational cost.

**Questions:**

1. Block diffusion models inherently have the potential to reuse KV-cache in multi-turn dialogues. However, E2D2 introduces an encoder that must process the entire history, which seems to break the ability to efficiently reuse KV-cache in long conversations. It will be better to  provide a detailed explanation of E2D2's scalability in multi-turn, long-dialogue scenarios and compare it to standard block diffusion models.

2. None of the results tables in the paper clearly denote which specific model size was used (0.6B or 1.7B). For the sake of reproducibility and fair assessment, it is important to provide this information for all tables.

3. The authors may need to  provide a thorough analysis explaining why their model's performance on GSM-8K dropped by over 40 percentage points compared to its powerful pre-trained base model. Is it using diiferent evaluation settings?

If the authors can solve my questions, I may consider to raise my score.

**Ethical Concerns:**

["NO or VERY MINOR ethics concerns only"]

**Final Justification:**

4: Borderline accept

**Limitations:**

See weakness and questions.

**Quality:**

2

**Strengths And Weaknesses:**

Strengths:
1. The core strength is the proposal of E2D2, which addresses the weaknesses between autoregressive models and non-autoregressive models like standard diffusion. The hybrid design retains the sequential logic of AR models at a macro level between blocks while harnessing the parallel efficiency of diffusion at a micro level within each block.
2. The paper provides experiments on different benchmarks to prove the dominant performance, the strong throughput and the execellent generalization.

Weakness:
1. The core contribution is not clear enough. The proposed method appears to be merely a straightforward combination of block diffusion like BD3-LMs[1] and the encoder-decoder architecture like T5[2], rendering the limited contribution.
2. The paper lacks the necessary discussion of some related works. For example, the authors failed to cite or compare their work with a highly relevant NLP paper on multi-token prediction[3], next block generation[4].
3. Though the authors state their model is fine-tuned from Qwen3-1.7B. It shows that the original Qwen3-1.7B base model achieves an accuracy of 75.44% on the GSM-8K benchmark. However, E2D2 only reports an accuracy of 33.89% in Table 7. Given that the base model's training set already includes GSM-8K, such a performance drop is  incomprehensibl.
4. In Table 5, E2D2 shows only a little advantage over its direct counterpart, the BD3-LM baseline, which weakens the value of its contribution.

---

> ### Author Rebuttal · Authors · 2025-07-31
>
> We thank the reviewer for their feedback. Please see below for responses to the specific criticisms and questions raised.
>
> ---
>
> ### **Concern 1:** Limited novelty
>
> We argue that our work has novelty beyond combining a T5-style architecture with BD3-LM. Specifically, it involves the following research challenges:
> - **Architecture Design.** There are multiple ways to define encoder-decoder inputs and outputs. See e.g., ENAT [5] for an alternative architecture optimized for images. More generally, we report ablations on ways of connecting the encoder and decoder, the interplay of block size and decoder size, choices of decoder optimizations, weight-tying, and more.
> - **Training Algorithms.** Our architecture requires specialized training algorithms that extend BD3-LM (Algorithm 1), which rely on custom attention masks relative to BD3LM (Figure 2) and a modified computation graph (Figure 1) relative to T5.
> - **Sampling Algorithms.** Similarly, we introduce sampling algorithms that extend BD3-LM (Algorithm 2) and provide hyper-parameters (e.g., block size, number of layers) needed to obtain good performance.
>
> ---
>
> ### **Concern 2:** Clarifying why fine-tuned models have drop in performance relative to base model
>
> There are several factors at play here. First, although E2D2 is within 1 perplexity point of the base Qwen model [1], this gap translates to a larger drop in GSM8K exact-match accuracy which is sensitive to small mistakes in the arithmetic trace. Qualitatively, we observe that even single-token mismatches may result in an incorrect response. Additionally, the gap between even the fine-tuned AR model and the value reported in the original Qwen3 technical report [1] stems from the different evaluation setup, as detailed below.
>
>
> **GSM8k Results**: (Best diffusion values are **bolded**)
>
> |     | Fine-tuned AR | E2D2 (Ours) |
> |---------------|---------|---------|
> | PPL ($\downarrow$) | 1.52    | 67.6 |
> | pass @ 1 accuracy  ($\uparrow$)  | 1.77  | 39.73 |
>
> (Note: the E2D2 values here represent an improved fine-tuning setup with smaller batch size, which we will update in our revised manuscript).
>
> In terms of the differences in evaluation setup:
> - The original Qwen model used 4-shot CoT reasoning traces. Our evaluation is 0-shot.
> - Additionally, we fine-tuned the model to just output “reasoning traces” provided in the GSM8k dataset. This differs significantly from the default output that the Qwen model produces which is much more verbose. We limited our scope of number of shots and reasoning traces for computational considerations.
>
> We also comment that this gap is limited to the GSM8k dataset. On the translation and summarization tasks, however, E2D2 _beats_ the performance of the fine-tuned AR model (Tables 5 & 6):
>
> |     | Fine-tuned AR | E2D2 (Ours) |
> |---------------|---------|---------|
> | WMT BLEU ($\uparrow$) |  21.45    | 21.51 |
> | CNN/DM ROUGE-1 ($\uparrow$) | 26.4  | 36.8 |
>
> ---
>
> ### **Concern 3:** Limited improvement of E2D2 vs. BD3LM
>
> While performance relative to BD3LM is comparable in Table 5, we note that we achieve this at either improved efficiency (relative to the train-matched BD3LM) or no additional cost (relative to the inference matched BD3LM).
>
> To make this tradeoff of efficiency and performance more concrete, we provide a table showing the Pareto frontier between model performance (measured by PPL and pass@1) and throughput for E2D2 vs. BD3LM models. We find that **for comparable throughput, E2D2 consistently features higher quality.** Thus, it extends forward the Pareto frontier of quality and speed. We obtain best results for high and medium throughput levels, where the gain in quality is especially high.
>
> To measure this frontier we vary model size (number of layers) which controls both the performance (larger model $\rightarrow$ higher performance) and throughput (larger model $\rightarrow$ lower throughput) axes.
>
> | Model | # Layers | PPL ($\downarrow$) | 0-shot pass @1 ($\uparrow$) | Inference Throughput (Tok/sec $\uparrow$) |
> |--|--|--|--|--|
> | _High throughput_ |
> | BD3LM | 17 | 2.16 | 15.39 | 99.54 $\pm$ 1.13 |
> | E2D2 | 28 enc / 14 dec | **1.77** | **39.73** | 99.02 $\pm$ 1.45 |
> | _Medium throughput_ |
> | BD3LM | 21 | 1.86 | 30.25 | **83.57** $\pm$ 1.20 |
> | E2D2 | 28 enc / 21 dec | 1.73 | **43.29** | 78.65 $\pm$ 0.94 |
> | _Low throughput_ |
> | BD3LM | 28 | 1.65 | 44.73 | 65.93 $\pm$ 0.75 |
> | E2D2 | 28 enc / 24 dec | 1.72 | **46.17** | **71.81** $\pm$ 0.75 |
>
> **Exp. details:**
> - We fine-tuned Qwen 1.7B-Base models on the gsm8k train dataset.
> - Models were fine-tuned with batch size of 1, for 30k steps, which we found to be a more useful setup than the one currently reported in the manuscript. We will update Table 7 with these[ improved numbers .
> - Throughput calculation was done using batch size of 1 on a single H100 GPU. Models generated 256 new tokens for 50 prompts from the gsm8k test dataset. We report mean $\pm$ standard deviation across samples. Note also that the throughput values reported here are superior to those in the original manuscript as we found inefficiencies in our code that we were able to streamline.
> - For MDLM, we generate semi-autoregressively using blocks of 64, to have a fair comparison to the other models. We did a sweep on block sizes from 4 to 256 for MDLM to find the best performing one.
> - Note, E2D2 has the same trainable parameters as AR and MDLM since we tie the weights of the models here.
>
> ---
>
> ### **Concern 4:** Missing related works
>
> Thank you for these suggested related works. We are happy to include them in our revised manuscript. Unfortunately, the exact references for “[3]” and “[4]” suggested by the reviewer appear to have been cut off in the OpenReview UI. We therefore present discussions for what we assume are the works to which the reviewer was referring. Please let us know if we are not discussing the ones you had in mind:
>
> **“multi-token prediction [3]”**: _Gloeckle, Fabian, et al. "Better & faster large language models via multi-token prediction."  (2024)._
>
> Our work differs from [3] in several important ways. First, for their non-parallel generation, within a block, [3] maintains an autoregressive decoding scheme. In contrast, our diffusion parameterization within each block enables any-order generation. Additionally, similar to our contrast with works, such as Medusa [5], our architecture represents a more expressive decoder relative to the “heads” used in [3] and [5] and E2D2 has  more sophisticated information sharing between the encoder and the decoder via cross-attention to the encoder outputs.
>
> **“next block generation [4]”** _Ho, Namgyu, et al. "Block transformer: Global-to-local language modeling for fast inference." (2024)_
>
> The key difference between E2D2 and this work are as follows: first [4] is still parameterized as an AR model, whereas E2D2 uses a block diffusion parameterization. Additionally, E2D2 employs standard global, self-attention whereas the efficiencies from [4] come from a custom hierarchical transformer model that first pools tokens to create block embeddings over which global attention is applied and then performs only local attention within blocks to decode autoregressively. These efficiencies are orthogonal to our proposed method and in fact could potentially be combined with E2D2 by replacing the standard transformer with the Block Transformer, to reduce KV cache memory storage and retrieval bottlenecks.
>
> ---
>
> ### **Question 1:** Can E2D2’s encoder support efficient KV caching
>
> We clarify that E2D2 also supports efficient caching. Both the encoder and decoder use a KV cache of the already decoded / clean context blocks. More specifically, in E2D2, the encoder first processes clean context tokens. These are stored in a KV cache for the encoder (which the decoder also uses). Each newly decoded block is added to this KV cache in the same manner they would for BD3LM, except in E2D2, this is done by processing them in the encoder.
>
> Thus KV-cache is not a bottleneck for E2D2 models in scaling to multi-turn, long-dialogue settings.
>
> ---
>
> ### **Question 2:** Clarifying the model sizes used in each experiment
>
> Thank you for the suggestion. This information (reproduced in the table below) is included in the appendix, but to improve readability we will add it to the main text and corresponding tables as well.
>
> | Experiment | Base model size |
> |---|---|
> | Machine translation (WMT) | Qwen/Qwen3-0.6B-Base |
> | Summarization (CNN-DailyMail) | Qwen/Qwen3-0.6B-Base |
> | Math reasoning (GSM8k) | Qwen/Qwen3-1.7B-Base |
>
>
> ---
>
> **References**:
>
> [3]  Gloeckle, Fabian, et al. "Better & faster large language models via multi-token prediction." arXiv preprint arXiv:2404.19737 (2024).
>
> [4] Ho, Namgyu, et al. "Block transformer: Global-to-local language modeling for fast inference." Advances in Neural Information Processing Systems 37 (2024): 48740-48783.
>
> [5] Cai, Tianle, et al. "Medusa: Simple llm inference acceleration framework with multiple decoding heads." arXiv preprint arXiv:2401.10774 (2024).

---

> > ### Comment · Reviewer_xEvo · 2025-08-04
> >
> > Thanks for the hard work. The rebuttal almost solves my questions. However, regarding Concern 2, if the performance drop on GSM-8K is due to different evaluation setups, it would be helpful to run an additional comparison under the same evaluation setting. Specifically, could you compare the performance of your fine-tuned E2D2 and fine-tuned AR under the same setup (including both 0-shot and 4-shot, since both are reported in the paper, or you can just choose one of them)? This would clarify whether the performance gap is truly due to evaluation differences or model capability.

---

> > > ### Author Response · Authors · 2025-08-05
> > >
> > > Thank you for the continued engagement with our work and apologies for the remaining confusion. Our comment regarding the differing eval setups referred to the original Qwen3 technical report (4shot) vs. our work (0shot). Let us clarify that **all** of the evals in our manuscript were done 0-shot. That is, the **fine-tuned AR and E2D2 in our work have the exact same evaluation setup**. Below we re-paste the 0shot eval results for fine-tuned AR vs E2D2:
> > >
> > > **Table:** _0-shot evaluation of fine-tuned GSM8k models_
> > >
> > > |     | Fine-tuned AR | E2D2 (Ours)
> > > |---------------|---------|---------|
> > > | PPL ($\downarrow$) | 1.52    | 67.6 |
> > > | 0shot pass @ 1 accuracy  ($\uparrow$)  | 1.77  | 39.73 |
> > >
> > >
> > > Thus there does indeed exist a persistent downstream performance gap between AR and our diffusion-based models on GSM8k. As mentioned in our previous reply, this gap is driven by the fact  GSM8k is **highly** sensitive to small decoding errors and the small gap in perplexity between the fine-tuned AR model and E2D2 model is exaggerated in terms of pass @ 1 accuracy. We note that for other downstream tasks, such as translation and summarization, where the metrics (e.g., BLEU, ROUGE) are more ‘forgiving’, this downstream performance gap between fine-tuned E2D2 and AR does not exist.
> > >
> > > Please do not hesitate to let us know if anything else remains unclear about this evaluation or the other discussion points from our rebuttal.
> > >
> > > We look forward to your response.

---

> > > > ### Comment · Reviewer_xEvo · 2025-08-06
> > > >
> > > > Thanks for the rebuttal. I will raise my score to 4

---

> > > > > ### Author Response · Authors · 2025-08-06
> > > > >
> > > > > Thank you very much for the valuable feedback and discussion.

---

> ### Author Response · Authors · 2025-08-04
>
> Dear Reviewer,
>
> Thank you again for providing your initial feedback on our work. Given that the discussion period ends in two days, we wanted to reach out and see if there were any additional points of clarification we can provide. Did our rebuttal address your concerns and questions? If so, would you kindly be willing to reconsider your initial score of our work?
>
> Looking forward to your feedback.
>
>
> Sincerely,
>
> The Authors

---

### Official Review · Reviewer_6TKj · 2025-06-28

**Clarity:** 4
**Significance:** 3
**Originality:** 3
**Rating:** 4
**Confidence:** 4

**Summary:**

This paper introduces E2D2, an extension to block diffusion language models that decouples the size of the decoder from the encoder, improving performance for fixed training/inference speeds. The core idea in the paper is that the encoder context of block diffusion models—corresponding to the cross attention of the diffusing block and prior context—can be differentiated from the decoding model that generates the diffusion predictions. This equates to an encoder-decoder framework and opens up a number of avenues for exploration. Specifically, E2D2 proposes removing a number of the layers closer to the input so that the decoder is smaller than the encoder. This has the effect of either improving the cost of training as the noised examples go through fewer layers of the architecture. Alternatively, the model size can be modified to keep inference costs the same allowing for a larger model to create the KV cache with amortized cost followed by repeated steps of the diffusion model. The authors then explore a number of design decisions for the E2D2 models including weight-tying and fine-tuning datasets, before providing analyzing the results of the E2D2 framework on a variety of benchmarks such as ROUGE scores, WMT (de-en) BLEU scores, and throughput-comparisons highlighting the benefits of the E2D2 framework.

**Questions:**

Questions:
1) Is the use of a KV-cache novel to this work? It is my understanding the the BD3LM paper uses a clean input processed to also generate a KV cache for further decoding. Could you expand on the differences between the methods?
2) Related to the first limitation, but could the smaller diffusion models make more sense the larger the encoder network is?
3) How were the “inference-matched” and “train-matched” baselines chosen? The pruning performed by choosing the first-N layers of a model seems unclear if that would be the most performant result to explore?

Nit:
1) There is some inconsistencies in the naming for the Tables, e.g. (“inf_match” vs. “inference-matched”, consistency would be great).
2) Including the direction of improvement for Figures and Tables would be helpful for readability
3) It is worth highlighting in the main body of the paper that a block size of 4 was used for most experiments.

**Ethical Concerns:**

["NO or VERY MINOR ethics concerns only"]

**Final Justification:**

I would upgrade my recommended score to a 4. Though the original paper was missing a number of key results surrounding the pareto frontier of results, I believe that the results provided in the rebuttal (when included in the final version of the paper) will make for a good paper and incite discussion into the encoder-decoder framing of block discrete diffusion models.

Additional points considered and answered:
1) The added point about the design of the decoder framework (for example, the 4 layer decoder presented in the rebuttal or using the last layer hidden representations) helps complete the discussion around network design.
2) The WMT experiment also helps portray a complete story of training from scratch and shows that network design is not conflated by other sources of training instability.

**Limitations:**

Yes

**Paper Formatting Concerns:**

Not applicable - see nits above

**Quality:**

2

**Strengths And Weaknesses:**

Strengths:
1) The paper highlights a core attribute of the block diffusion language model that size of the encoder and decoder can be decoupled. This allows for greater flexibility in terms of model design, in this case in terms of the number of layers of the diffusion model.
2) There are a number of interesting ablations that are helpful for practitioners aiming to use E2D2 and resulting frameworks. For example, the tied weights is promising for limiting the memory constraints of the resulting model. The throughput comparisons may also help researchers choose what size of E2D2 model to use.

Limitations:
1) While enabling a new paradigm for decoupling the encoder-and-decoder components of the language model, there is a significant amount of flexibility in the distinctions between the two models that remains unexplored in this paper. A number of immediate ablations, such as how many layers are necessary in the decoder network are missing, limiting the conclusions that can be drawn. The paper could benefit from fully exploring the Pareto frontier of decoder-size, as it appears that the decoder could even be shrunk further. While there is some discussion of this in the appendix (with the number of decoder layers), the choice of n/2 in the main body of the paper appears limited and does not explore the range of possible options. For example, another option would be to maintain depth but shrink the model size of the decoder, use grouped attention / similar inference optimizations / alternatively modify the decoder in a way that decouples the behavior. Another direction for exploration is whether the size of the decoder affects the number of steps that the diffusion model needs to perform for a fixed target perplexity.
2) The paper also mostly focuses on fine-tuning datasets, which is interesting in itself. However, the limited results from “From scratch” training draw concern to if the method leads to training instability or similar that would prevent training a diffusion language model from scratch with this method.

---

> ### Author Rebuttal · Authors · 2025-07-31
>
> We thank the reviewer for their valuable feedback and for recognizing the contributions of our work. Below we respond to the specific concerns and questions
>
> ---
>
> ### **Concern 1:** Provide a more complete picture of the “pareto frontier”
>
> Below, we provide a table showing the Pareto frontier between model performance (measured by PPL and pass@1) and throughput for E2D2 vs. BD3LM models. We find that **for comparable throughput, E2D2 consistently features higher quality.** Thus, it extends forward the Pareto frontier of quality and speed. We obtain best results for high and medium throughput levels, where the gain in quality is especially high.
>
> To measure this frontier we vary model size (number of layers) which controls both the performance (larger model $\rightarrow$ higher performance) and throughput (larger model $\rightarrow$ lower throughput) axes.
>
> | Model | # Layers | PPL ($\downarrow$) | 0-shot pass @1 ($\uparrow$) | Inference Throughput (Tok/sec $\uparrow$) |
> |--|--|--|--|--|
> | _High throughput_ |
> | BD3LM | 17 | 2.16 | 15.39 | 99.54 $\pm$ 1.13 |
> | E2D2 | 28 enc / 14 dec | 1.77 | **39.73** | 99.02 $\pm$ 1.45 |
> | _Medium throughput_ |
> | BD3LM | 21 | 1.86 | 30.25 | **83.57** $\pm$ 1.20 |
> | E2D2 | 28 enc / 21 dec | 1.73 | **43.29** | 78.65 $\pm$ 0.94 |
> | _Low throughput_ |
> | BD3LM | 28 | 1.65 | 44.73 | 65.93 $\pm$ 0.75 |
> | E2D2 | 28 enc / 24 dec | 1.72 | **46.17** | **71.81** $\pm$ 0.75 |
>
> **Exp. details:**
> - Best values beyond margin of error are **bolded**
> - We fine-tuned Qwen 1.7B-Base models on the gsm8k train dataset.
> - Models were fine-tuned with batch size of 1, for 30k steps, which we found to be a more useful setup than the one currently reported in the manuscript. We will update Table 7 with these[ improved numbers .
> - Throughput calculation was done using batch size of 1 on a single H100 GPU. Models generated 256 new tokens for 50 prompts from the gsm8k test dataset. We report mean $\pm$ standard deviation across samples. Note also that the throughput values reported here are superior to those in the original manuscript as we found inefficiencies in our code that we were able to streamline.
> - For MDLM, we generate semi-autoregressively using blocks of 64, to have a fair comparison to the other models. We did a sweep on block sizes from 4 to 256 for MDLM to find the best performing one.
> - Note, E2D2 has the same trainable parameters as AR and MDLM since we tie the weights of the models here.
>
> ---
>
> ### **Concern 2:** Explore other design choices for decoupling Encoder-Decoder architecture design choices are underexplored
>
> Here, we explore an alternative way of connecting the encoder and decoder: rather than sharing KV cache for matched layers in the encoder decoder, we feed the last hidden layer representation of the encoder into the decoder. We conducted an experiment for this new model by training from scratch on the machine translation (WMT14 de-en) dataset. In this setup, we also find that **E2D2 is more efficient and better performing** compared to MDLM and BD3LM.
>
> **WMT results:**
> | Model | # Layers | # Params | PPL ($\downarrow$) | BLEU ($\uparrow$) | Inference Throughput (Tok/sec $\uparrow$) |
> |--|--|--|--|--|--|
> | MDLM | 32 | 254M | 6.80 | 20.11 | 55.95 $\pm$ 8.80  |
> | BD3LM | 12 | 144M  | 7.92 | 23.71 | 81.02 $\pm$ 1.10 |
> | BD3LM | 16 |166M | 7.36 | 24.11 | 63.73 $\pm$ 0.66 |
> | E2D2 | 28 enc / 4 dec | 254 M | **6.49** | **25.01**  | **101.21** $\pm$ 3.91 |
>
> Although E2D2 has more parameters than the BD3LM baseline, it is both faster and has better downstream performance, highlighting the benefit of our proposed encoder-decoder structure.
>
> **WMT Exp. details:**
> - We train models with hidden dimension 512 and MLP dimension of 1536 **from scratch**. Other model architecture details follow Qwen3 models.
> - Batch size of 128, LR 3e-4, trained for 1M steps. We perform early stopping using eval loss.
> - Throughput calculation was done using batch size of 1 on a single H100 GPU. Models generated 256 new tokens for 50 prompts from the test dataset. We report mean $\pm$ standard deviation across samples.
> - For MDLM, we generate semi-autoregressively using blocks of 32, to have a fair comparison to the other models. We did a sweep on block sizes from 4 to 256 for MDLM to find the best performing one.
> - **Note**: We use a modified architecture for E2D2 here: encoder and decoder weights are untied. Rather than sharing KV cache between the encoder and decoder as above, we pass the last hidden representation from the encoder to each layer of the decoder.
>
>
> Next, we address this comment from the reviewer:
> > Another direction for exploration is whether the size of the decoder affects the number of steps that the diffusion model needs to perform for a fixed target perplexity.
>
> Below we present an experiment that compares varying the number of diffusion steps per block for E2D2, across different decoder sizes. The efficiency of decoupling encoder and decoder compute in E2D2 allows us to use larger models with fewer decoding steps and still outperform BD3LM models (that use more diffusion steps) in terms of both the throughput and benchmark performance axes.
>
> | Model   | Layers   |   T | Inference Throughput (Tok/sec $\uparrow$)   |   0-shot pass @1 ($\uparrow$) |
> |---|---|---|---|---|
> | BD3LM   | 17 |   4 | 99.54 $\pm$ 1.13 | 15.39 |
> | BD3LM   | 21|   4 | 83.57 $\pm$ 1.20 | 29.42 |
> | E2D2    | 28 enc / 14 dec |   2 | 152.05 $\pm$ 3.24| 30.17 |
> | E2D2    | 28 enc / 21 dec |   2 | 126.57 $\pm$ 2.13| 32.83 |
> | E2D2    | 28 enc / 24 dec |   2 | 119.50 $\pm$ 1.64| 35.33 |
>
> (**Exp. details:** Same as above)
>
> ---
>
> ### **Concern 3:** Add a “from scratch” experiment
>
> This comment is well received. The WMT experiment from above was indeed conducted from scratch and demonstrates that our approach is not reliant on initializing from pre trained models.
>
> ---
>
> ### **Formatting concerns:**
>
> Thank you for these comments. We will update the revised manuscript accordingly.
>
> ---
>
> ### **Question 1:** “Is the use of a KV-cache novel to this work?”
>
> To clarify, both E2D2 and BD3LM enable KV-caching of already decoded blocks and this is one of the primary motivations of building E2D2 on BD3LM models as opposed to their predecessor vanilla MDLM models, which do not support KV caching (in addition to the fact that BD3LM boasts improved performance vs. MDLM). The difference between these models is that to decode a given token one needs to make a pass through all the layers of the BD3LM model, whereas for E2D2 we only require a forward pass through the smaller number of decoder layers.
>
> Thus, for the sake of illustration, assume that the E2D2 encoder is the same size as the full BD3LM model, then the KV cache for both BD3LM and E2D2 would be the same: cache any context/prompt tokens and, as each new block is generated, add it to the cache (by running it through the full network for BD3LM or through the encoder for E2D2). For E2D2, the decoder uses the KV cache from the encoder as well.
>
> ---
>
> ### **Question 2:** “Could the smaller diffusion models make more sense the larger the encoder network is?”
>
> Great question. We explore this above in the response to concern 1.
>
> ---
>
> ### **Question 3:** “How were the ‘inference-matched’ and ‘train-matched’ baselines chosen?”
>
> These baselines were chosen based on the FLOPs computations provided in Tables 1 and 2.
> - **Train-matched:** To match the training flops between E2D2 and BD3LM, the BD3LM model needs to have half as many layers as the combined number of encoder and decoder layers of the E2D2 model. See Table 1.
> - **Inference-matched:** We matched the inference compute empirically by seeing the number of layers of a BD3LM model that would lead to similar throughput to an E2D2 model (assuming we only decode one token per forward pass of the diffusion model, i.e. diffusion time steps $T$ per block is equal to block size $S$). See the throughput results from the response to Concern 1 above.

---

> ### Author Response · Authors · 2025-08-04
>
> Dear Reviewer,
>
> Thank you again for providing your initial feedback on our work. Given that the discussion period ends in two days, we wanted to reach out and see if there were any additional points of clarification we can provide. Did our rebuttal address your concerns and questions? If so, would you kindly be willing to reconsider your initial score of our work?
>
> Looking forward to your feedback.
>
>
> Sincerely,
>
> The Authors

---

> > ### Comment · Reviewer_6TKj · 2025-08-05
> > **Discussion**
> >
> > Thank you for the comments. Apologies for the delay in response.
> >
> > The rebuttal mostly resolves my concerns with the paper. I think the clearer description of the pareto frontier really helps highlight how this strategy can be used to combine autoregressive and diffusion generation. In addition, the experiments around the alternative strategy for sharing the KV cache from the last layer are convincing that there is significant flexibility in the decoupled architecture.
> >
> > I am more supportive of this work and hope that these additional details are added to the final body of the paper.

---

> ### Author Response · Authors · 2025-08-06
>
> Thank you again for the thoughtful feedback and discussion, which were instrumental in helping us gather and better frame these new results. The new results will certainly be included in our updated manuscript.
>
> We hope that this positive and productive discussion period will be reflected in your final assessment of our work.
>
>
> Sincerely,
>
> The Authors

---

### Official Review · Reviewer_4wvn · 2025-06-30

**Clarity:** 3
**Significance:** 2
**Originality:** 3
**Rating:** 5
**Confidence:** 3

**Summary:**

The authors introduce E2D2, an encoder-decoder method for discrete diffusion. The method uses the encoder to create representations of the input that the decoder cross-attends to the Key and Value outputs of the input. The authors then train 3 domain specific models for summarization (CNN/DailyMail), translation (WMT), and math reasoning (GSM8K). Their experiments show a performance increase with E2D2 to a modified BD3LM.

**Questions:**

Can you share the results of an unmodified BD3LM on the tasks presented in the paper? As well as the inference and training throughput?

How is the AR model trained / what does the model size look like? Did you inference match / train match the AR model?

Are the experiments in 4.4 fine tuned individually (i.e., for each parameter) or the results with the same base model?

**Ethical Concerns:**

["NO or VERY MINOR ethics concerns only"]

**Final Justification:**

The author responses alleviate concerns and increase clarity. The Pareto results added during the response makes further clear the benefits of this work. I have increased my score accordingly.

**Limitations:**

I have no suggestions on discussing any negative societal impact of this particular work

**Quality:**

3

**Strengths And Weaknesses:**

Strengths
The application of Encoder-Decoder models to Diffusion LMs is interesting.

The ablation experiments are interesting showing the tradeoffs between performance and throughput of the E2D2 method.


Weakness
The comparisons to BD3LM could be expanded. Currently the comparison is “apples-to-apples” wrt compute, but it would also be valuable to see the performance differences and efficiency of an unmodified model. The authors refer to a “Performance/Throughput Pareto frontier”, but it is not then obvious where E2D2 sits compared to other existing models when provided larger compute budgets.

Extra result:
I am unclear on the exact evaluation setting of these experiments (see the questions section), but the block size experiments could be potentially very interesting. It would be interesting to also see the summarization and translation experiments for this setting.

---

> ### Author Rebuttal · Authors · 2025-07-31
>
> We thank the reviewer for their feedback and for the valuable additional experiment suggestions. Below we respond to the specific comments inline.
>
> ---
>
> ### **Concern 1:** Provide a more complete “pareto frontier” picture
>
> Per the reviewer request, below, we provide a table showing the Pareto frontier between model performance (measured by PPL and pass@1) and throughput for E2D2 vs. BD3LM models. We find that **for comparable throughput, E2D2 consistently features higher quality.** Thus, it extends forward the Pareto frontier of quality and speed. We obtain best results for high and medium throughput levels, where the gain in quality is especially high.
>
> To measure this frontier we vary model size (number of layers) which controls both the performance (larger model $\rightarrow$ higher performance) and throughput (larger model $\rightarrow$ lower throughput) axes.
>
> (Note: This table also address the reviewer’s request for presenting the  “unmodified” BD3LM, i.e., the 28 layer BD3LM).
>
> | Model | # Layers | PPL ($\downarrow$) | 0-shot pass @1 ($\uparrow$) | Inference Throughput (Tok/sec $\uparrow$) |
> |--|--|--|--|--|
> | _High throughput_ |
> | BD3LM | 17 | 2.16 | 15.39 | 99.54 $\pm$ 1.13 |
> | E2D2 | 28 enc / 14 dec | 1.77 | **39.73** | 99.02 $\pm$ 1.45 |
> | _Medium throughput_ |
> | BD3LM | 21 | 1.86 | 30.25 | **83.57** $\pm$ 1.20 |
> | E2D2 | 28 enc / 21 dec | 1.73 | **43.29** | 78.65 $\pm$ 0.94 |
> | _Low throughput_ |
> | BD3LM | 28 | 1.65 | 44.73 | 65.93 $\pm$ 0.75 |
> | E2D2 | 28 enc / 24 dec | 1.72 | **46.17** | **71.81** $\pm$ 0.75 |
>
> **Exp. details:**
> - Best values beyond margin of error are **bolded**
> - We fine-tuned Qwen 1.7B-Base models on the gsm8k train dataset.
> - Models were fine-tuned with batch size of 1, for 30k steps, which we found to be a more useful setup than the one currently reported in the manuscript. We will update Table 7 with these[ improved numbers .
> - Throughput calculation was done using batch size of 1 on a single H100 GPU. Models generated 256 new tokens for 50 prompts from the gsm8k test dataset. We report mean $\pm$ standard deviation across samples. Note also that the throughput values reported here are superior to those in the original manuscript as we found inefficiencies in our code that we were able to streamline.
> - For MDLM, we generate semi-autoregressively using blocks of 64, to have a fair comparison to the other models. We did a sweep on block sizes from 4 to 256 for MDLM to find the best performing one.
> - Note, E2D2 has the same trainable parameters as AR and MDLM since we tie the weights of the models here.
>
> ---
>
> ### **Concern 2:** Explore the effect of block size more
>
> Below we present the exploration of the effect of block size for E2D2 models. We find that E2D2 with block size of either 4 or 8 better trades-off benchmark performance with throughput.
>
> | Block size | 0-shot pass @1 ($\uparrow$) | Inference Throughput (Tok/sec $\uparrow$) |
> |--|--|--|
> | 4 |  39.73 | 99.02 $\pm$ 1.45 |
> | 8 |  34.42 | 108.58 $\pm$ 1.75 |
> | 16 |  30.48 | 103.70 $\pm$ 1.48 |
>
> Note that all models above were fine-tuned as separate runs (see the answer to the related question below). This is an updated version of Table 10 from our paper which we will revise in our latest manuscript.
>
> **Exp details**: Same as above.
>
> ---
>
> ### **Question 1:** How is the AR model trained / what does the model size look like?
>
> All of the models are based on the same backbone and are hence comparable. The AR model in particular represents an unmodified backbone, that is we use all 28 layers of the pre-trained model. This is akin to matching the number of encoder layers of E2D2. This renders the AR model a very strong baseline. We will add this information more explicitly in our revised manuscript
>
> ---
>
> ### **Question 2:** Are the experiments in 4.4 fine tuned individually (i.e., for each parameter) or the results with the same base model?
>
> We apologize for the confusion. In our updated manuscript we will make it clear that each result in the ablations section (4.4) for both Table 9 and Table 10 represents a separate fine-tuning run. That is, for each row in these tables, starting from the same baseline pre-trained model, we initialize weights from the pre-trained model, modify the number or layers (Table 9) or block size (Table 10), and then fine-tune. This holds true for the updated results presented in this response as well, which we will use when revising our manuscript.

---

> ### Author Response · Authors · 2025-08-04
>
> Dear Reviewer,
>
> Thank you again for providing your initial feedback on our work. Given that the discussion period ends in two days, we wanted to reach out and see if there were any additional points of clarification we can provide. Did our rebuttal address your concerns and questions? If so, would you kindly be willing to reconsider your initial score of our work?
>
> Looking forward to your feedback.
>
>
> Sincerely,
>
> The Authors

---

### Official Review · Reviewer_KkBK · 2025-07-01

**Clarity:** 3
**Significance:** 2
**Originality:** 3
**Rating:** 4
**Confidence:** 4

**Summary:**

This paper proposes an encoder-decoder architecture for block discrete diffusion models, which is called E2D2, addressing the doubled computational costs inherent in standard decoder-only approaches. By employing a lightweight decoder that iteratively refines each generated block, the method achieves improved efficiency while maintaining generation quality. Furthermore, the experiments demonstrate the effectiveness of E2D2, showing that E2D2 achieves faster inference speed and lower training costs compared to the standard decoder-only approach.

**Questions:**

- Regarding weakness 2, how can the optimal settings for the number of decoder layers and block size be determined?
- In Table 4, the training and decoding speeds of the AR model are superior to those of the block diffusion models. Could the authors elaborate on this result and clarify the primary advantages of block diffusion models over AR models?

**Ethical Concerns:**

["NO or VERY MINOR ethics concerns only"]

**Final Justification:**

After rebuttal, the authors address all my concern and i increase my score to 4.

**Limitations:**

The paper does not include a limitations or potential negative societal impact section. I recommend adding a discussion on the performance gap that still exists compared to AR models and the challenge of effectively selecting hyperparameters like the number of decoder layers and block size.

**Paper Formatting Concerns:**

No.

**Quality:**

2

**Strengths And Weaknesses:**

**Strengths**
- The paper is well-structured and easy to understand.
- The proposed model is efficient in training and sampling, enabling faster inference and lower training costs compared to decoder-only block diffusion models.
- The paper provides a thorough comparison with standard decoder-only block diffusion models, demonstrating that the proposed model matches baseline performance.

**Weaknesses**
- The placement of tables and figures could be improved. For instance, Table 4 appears after Tables 5 and 6, and its embedding within the text is unconventional.
- Based on Tables 9 and 10, the model's decoding speed and performance appear sensitive to the number of decoder layers and the block size.
- The related work section omits a discussion of LLaDA, which is arguably the first large-scale diffusion language model and highly relevant to this work. Furthermore, several other non-autoregressive language models are not discussed, such as SEDD and MDLM.
- The paper lacks a performance comparison in table with other relevant discrete diffusion models, such as SEDD and MDLM.

**Minor Points:**
- There are minor typos. For instance, on line 39, 'Contributions' should be removed as it is not part of a sentence or a section title.
- Some phrasing is informal, such as 'we make three contributions' on line 39.

---

> ### Author Rebuttal · Authors · 2025-07-31
>
> We thank the reviewer for their feedback and for recognizing the contribution of E2D2 in improving discrete diffusion models. Below we respond to the reviewer’s concerns and questions.
>
> ### **Concern 1:** Missing comparison to models such as MDLM
>
> We thank the reviewer for this suggestion. Below we provide this comparison on the GSM8k dataset. We find that **MDLM significantly underperforms E2D2 on this task.**
>
> **GSM8k Results**: (Best diffusion values are **bolded**)
> | Model | # Layers | PPL ($\downarrow$) | 0-shot pass @1 ($\uparrow$) | Inference Throughput (Tok/sec $\uparrow$) |
> |--|--|--|--|--|
> | AR | 28 | 1.52 | 67.6 | 92.22 $\pm$ 1.36 |
> | MDLM | 28 | 2.34 | 14.33 | 31.35 $\pm$ 2.45 |
> | BD3LM | 21 | 1.86 | 30.25 | 83.57 $\pm$ 1.20 |
> | E2D2 | 28 enc / 14 dec | **1.77** | **39.73** | **99.02** $\pm$ 1.45 |
>
> We also provide this comparison for models trained **from scratch** on the machine translation (WMT de-en) dataset. Here too, **E2D2 is more efficient and better performing.**
>
> **WMT results:**
> | Model | # Layers | # Params | PPL ($\downarrow$) | BLEU ($\uparrow$) | Inference Throughput (Tok/sec $\uparrow$) |
> |--|--|--|--|--|--|
> | MDLM | 32 | 254M | 6.80 | 20.11 | 55.95 $\pm$ 8.80  |
> | BD3LM | 12 | 144M  | 7.92 | 23.71 | 81.02 $\pm$ 1.10 |
> | BD3LM | 16 |166M | 7.36 | 24.11 | 63.73 $\pm$ 0.66 |
> | E2D2 | 28 enc / 4 dec | 254 M | **6.49** | **25.01**  | **101.21** $\pm$ 3.91 |
>
> Although E2D2 has more parameters than the BD3LM baseline, it is both faster and has better downstream performance, highlighting the benefit of our proposed encoder-decoder structure.
>
>
> **GSM8k Exp. details:**
> - We fine-tuned Qwen 1.7B-Base models on the gsm8k train dataset.
> - Models were fine-tuned with batch size of 1, for 30k steps, which we found to be a more useful setup than the one currently reported in the manuscript. We will update Table 7 with these improved numbers .
> - Throughput calculation was done using batch size of 1 on a single H100 GPU. Models generated 256 new tokens for 50 prompts from the gsm8k test dataset. We report mean $\pm$ standard deviation across samples. Note also that the throughput values reported here are superior to those in the original manuscript as we found inefficiencies in our code that we were able to streamline.
> - For MDLM, we generate semi-autoregressively using blocks of 64, to have a fair comparison to the other models. We did a sweep on block sizes from 4 to 256 for MDLM to find the best performing one.
> - Note, E2D2 has the same trainable parameters as AR and MDLM since we tie the weights of the models here.
>
> **WMT Exp. details:**
> - We train models with hidden dimension 512 and MLP dimension of 1536 **from scratch**. Other model architecture details follow Qwen3 models.
> - Batch size of 128, LR 3e-4, trained for 1M steps. We perform early stopping using eval loss.
> - Throughput calculation was done using batch size of 1 on a single H100 GPU. Models generated 256 new tokens for 50 prompts from the test dataset. We report mean $\pm$ standard deviation across samples.
> - For MDLM, we generate semi-autoregressively using blocks of 32, to have a fair comparison to the other models. We did a sweep on block sizes from 4 to 256 for MDLM to find the best performing one.
> - **Note**: We use a modified architecture for E2D2 here: encoder and decoder weights are untied. Rather than sharing KV cache between the encoder and decoder as above, we pass the last hidden representation from the encoder to each layer of the decoder.
>
> ---
>
> ### **Concern 2:** Explaining the relationship between decoding throughput and number of decoder layers and block size
>
> As demonstrated in the asymptotic FLOPs computation of Tables 1 and 2, training and inference computational budgets are functions of both the number of layers and the block size. This is true for both E2D2 and BD3LM (and true for all models, e.g., AR and MDLM, in terms of the number of layers).
>
> In terms of the reviewer’s question on this topic:
> > Regarding weakness 2, how can the optimal settings for the number of decoder layers and block size be determined?
>
> The “optimal” architecture balances downstream performance and efficiency (i.e. throughput), which can be user-dependent. Models that better control this trade-off are preferable.
>
> Below, we provide a table showing the Pareto frontier between model performance (measured by PPL and pass@1) and throughput for E2D2 vs. BD3LM models. We find that **for comparable throughput, E2D2 consistently features higher quality.** Thus, it extends forward the Pareto frontier of quality and speed. We obtain best results for high and medium throughput levels, where the gain in quality is especially high.
>
> To measure this frontier we vary model size (number of layers) which controls both the performance (larger model $\rightarrow$ higher performance) and throughput (larger model $\rightarrow$ lower throughput) axes.
>
> | Model | # Layers | PPL ($\downarrow$) | 0-shot pass @1 ($\uparrow$) | Inference Throughput (Tok/sec $\uparrow$) |
> |--|--|--|--|--|
> | _High throughput_ |
> | BD3LM | 17 | 2.16 | 15.39 | 99.54 $\pm$ 1.13 |
> | E2D2 | 28 enc / 14 dec | **1.77** | **39.73** | 99.02 $\pm$ 1.45 |
> | _Medium throughput_ |
> | BD3LM | 21 | 1.86 | 30.25 | **83.57** $\pm$ 1.20 |
> | E2D2 | 28 enc / 21 dec | 1.73 | **43.29** | 78.65 $\pm$ 0.94 |
> | _Low throughput_ |
> | BD3LM | 28 | 1.65 | 44.73 | 65.93 $\pm$ 0.75 |
> | E2D2 | 28 enc / 24 dec | 1.72 | **46.17** | **71.81** $\pm$ 0.75 |
>
> (**Exp. details:** Same as above; Best values beyond margin of error are **bolded**)
>
> ---
>
> ### **Concern 3:** Missing discussion of Llada and predecessor models (MDLM)
>
> We thank the reviewer for this suggestion, and we will correct this oversight in the updated manuscript. Specifically we edit line 275 in the “Diffusion Language Models” paragraph of the related works as follows:
>
> > Recent large-scale diffusion language models have demonstrated competitive performance with comparably sized AR LLMs on benchmark metrics. Specifically, the Llada model [2], built on the work of  MDLM [3] Nie et al., 2024 [4], scales masked diffusion models to the 8B parameter regime and demonstrated comparable and superior performance to a similarly-sized AR Llama model [5].
>
> ---
>
> ### **Concern 4:** Missing “limitations” section
>
> Thank you for this suggestion. We will add the following text as a limitations section:
>
> > While our work represents a further step in closing the performance and efficiency gap to the dominant AR paradigm for language modeling, our experimental results demonstrate that discrete diffusion models still lag. Further innovation in training and inference algorithms for discrete diffusion models is needed. Additionally, since our work entails language modeling it carries the inherent risks of misuse of this technology. As these models improve in size and quality, care must be taken to prevent malicious use.
>
> ---
>
> ### **Additional concerns:** Table placement and typos
>
> Thank you for these suggestions. We will update the formatting and text in our revised manuscript.
>
> ---
>
> ### **Question:** Clarify the primary advantages of diffusion relative to AR, which has faster throughput
>
> As noted in the updated table above, we have removed an important inefficiency in our code which brings the throughput of the diffusion models in line with that of the AR baseline. Moveover, **we see the promise of E2D2 which is able to have the best diffusion performance with faster throughput than AR models**. We apologize that these results were not reflected in our original manuscript, but we will update them in our revised version.
>
> ---
>
> **References**
>
> [1] Arriola, Marianne, et al. “Block Diffusion: Interpolating Between Autoregressive and Diffusion Language Models.” ICLR 2025
>
> [2] Nie, Shen, et al. "Large language diffusion models." arXiv preprint arXiv:2502.09992 (2025).
>
> [3] Sahoo, Subham, et al. "Simple and effective masked diffusion language models." Advances in Neural Information Processing Systems 37 (2024): 130136-130184.
>
> [4] Nie, Shen, et al. "Scaling up masked diffusion models on text." arXiv preprint arXiv:2410.18514 (2024).
>
> [5] Dubey, Abhimanyu, et al. "The llama 3 herd of models." arXiv e-prints (2024): arXiv-2407.

---

> > ### Comment · Reviewer_KkBK · 2025-08-04
> >
> > Thank you for your response. I will take into consideration the comments from other reviewers before providing my final score.

---

> > > ### Author Response · Authors · 2025-08-04
> > >
> > > Thank you for your response. If there are any additional points of clarification or discussion please feel free to reach out.
> > >
> > > Sincerely,
> > > The Authors

---

> > > ### Author Response · Authors · 2025-08-09
> > > **Official Comment by Authors**
> > >
> > > We appreciate your response. As the other reviewers have updated their scores, we were wondering if their comments have led you to reconsider or adjust your assessment of our paper. Thank you!
> > >
> > > Sincerely, The Authors

---

> > > > ### Comment · Reviewer_KkBK · 2025-08-09
> > > >
> > > > Thank you for your response. I will raise my score to 4. However,  I believe the article could benefit from further refinement.

---

### Author Response · Authors · 2025-08-07
**Thank you and Rebuttal Summary**

Dear Reviewers,

Thank you again for your time and valuable feedback during the review process.

Below, we summarize the common feedback from the reviewers and the additional experimental results that we gathered during this process:

### **1. Mapped the Pareto frontier of efficiency and performance**

The discussion with several reviewers helped us better frame the benefit of E2D2. Specifically, we provided results showing the Pareto frontier between model performance (measured by PPL and pass@1) and throughput for E2D2 vs. BD3LM models. **For comparable throughput, E2D2 consistently features higher quality.**

| Model | # Layers | PPL ($\downarrow$) | 0-shot pass @1 ($\uparrow$) | Inference Throughput (Tok/sec $\uparrow$) |
|--|--|--|--|--|
| _High throughput_ |
| BD3LM | 17 | 2.16 | 15.39 | 99.54 $\pm$ 1.13 |
| E2D2 | 28 enc / 14 dec | **1.77** | **39.73** | 99.02 $\pm$ 1.45 |
| _Medium throughput_ |
| BD3LM | 21 | 1.86 | 30.25 | **83.57** $\pm$ 1.20 |
| E2D2 | 28 enc / 21 dec | 1.73 | **43.29** | 78.65 $\pm$ 0.94 |
| _Low throughput_ |
| BD3LM | 28 | 1.65 | 44.73 | 65.93 $\pm$ 0.75 |
| E2D2 | 28 enc / 24 dec | 1.72 | **46.17** | **71.81** $\pm$ 0.75 |

### **2. Added ‘from-scratch’ experiments and explored different architectures**

To demonstrate that our results do not rely on fine-tuning, we retrained models from scratch on the machine translation task (WMT). Moreover, we used this opportunity to explore new ways of connecting the modules in E2D2 (namely, letting the decoder cross-attend to the final layer output of the encoder). In this experiment, we again find that **E2D2 is more efficient and better performing.**

**WMT results:**
| Model | # Layers | # Params | PPL ($\downarrow$) | BLEU ($\uparrow$) | Inference Throughput (Tok/sec $\uparrow$) |
|--|--|--|--|--|--|
| MDLM | 32 | 254M | 6.80 | 20.11 | 55.95 $\pm$ 8.80  |
| BD3LM | 12 | 144M  | 7.92 | 23.71 | 81.02 $\pm$ 1.10 |
| BD3LM | 16 |166M | 7.36 | 24.11 | 63.73 $\pm$ 0.66 |
| E2D2 | 28 enc / 4 dec | 254 M | **6.49** | **25.01**  | **101.21** $\pm$ 3.91 |

### **3. Added baselines**

We added standard masked diffusion models (MDLM) as a baseline to our experiments. **E2D2 outperforms MDLM in terms of both downstream performance and throughput.**

**GSM8k Results**:
| Model | # Layers | PPL ($\downarrow$) | 0-shot pass @1 ($\uparrow$) | Inference Throughput (Tok/sec $\uparrow$) |
|--|--|--|--|--|
| MDLM | 28 | 2.34 | 14.33 | 31.35 $\pm$ 2.45 |
| E2D2 | 28 enc / 14 dec | **1.77** | **39.73** | **99.02** $\pm$ 1.45 |


**WMT results:**
| Model | # Layers | PPL ($\downarrow$) | BLEU ($\uparrow$) | Inference Throughput (Tok/sec $\uparrow$) |
|--|--|--|--|--|
| MDLM | 32 | 6.80 | 20.11 | 55.95 $\pm$ 8.80  |
| E2D2 | 28 enc / 4 dec | **6.49** | **25.01**  | **101.21** $\pm$ 3.91 |

---

Thank you again for your time and consideration.

Sincerely,

The Authors

---

### Decision · Program_Chairs · 2025-09-17

**Decision:**

Accept (poster)

**Comment:**

The paper introduces E2D2, an encoder–decoder discrete diffusion model that reduces the overhead of blockwise diffusion while maintaining or improving quality. It demonstrates strong efficiency–quality trade-offs across summarization, translation, and reasoning, marking a clear advancement over prior discrete diffusion approaches.
After rebuttal, I think the authors have addressed all the concerns of reviewers. And Reviewer xEvo said that she/he acknowledges the rebuttal and would like to raise the score to 4, so it seems all scores should be positive.